# Proliferative polyploid cells give rise to tumors via ploidy reduction

Tomonori Matsumoto [1✉], Leslie Wakefield[1], Alexander Peters[1], Myron Peto[2], Paul Spellman[2] & Markus Grompe [1✉]

Polyploidy is a hallmark of cancer, and closely related to chromosomal instability involved in cancer progression. Importantly, polyploid cells also exist in some normal tissues. Polyploid hepatocytes proliferate and dynamically reduce their ploidy during liver regeneration. This raises the question whether proliferating polyploids are prone to cancer via chromosome missegregation during mitosis and/or ploidy reduction. Conversely polyploids could be resistant to tumor development due to their redundant genomes. Therefore, the tumor-initiation risk of physiologic polyploidy and ploidy reduction is still unclear. Using in vivo lineage tracing we here show that polyploid hepatocytes readily form liver tumors via frequent ploidy reduction. Polyploid hepatocytes give rise to regenerative nodules with chromosome aberrations, which are enhanced by ploidy reduction. Although polyploidy should theoretically prevent tumor suppressor loss, the high frequency of ploidy reduction negates this protection. Importantly, polyploid hepatocytes that undergo multiple rounds of cell division become predominantly mononucleated and are resistant to ploidy reduction. Our results suggest that ploidy reduction is an early step in the initiation of carcinogenesis from polyploid hepatocytes.

[1] Department of Pediatrics, Oregon Health and Science University, Portland, OR, USA. [2] Department of Molecular and Medical Genetics, Oregon Health and Science University, Portland, OR, USA. ✉email: tomomatsumototm@gmail.com; grompem@ohsu.edu

Polyploidy is a characteristic alteration in tumors, and pan-cancer analyses showed that 28.2–37% of human cancers have undergone whole-genome duplication (polyploidization)[1,2]. Importantly, polyploidy is observed not only in cancers but also in premalignant lesions[3,4] and some normal cells such as hepatocytes[5]. Polyploidization is also enhanced in some disease conditions such as chronic liver injuries[6]. Given that mitosis of polyploid cells may be prone to induce aneuploidy via chromosome missegregation[3,7], proliferating polyploid cells in normal tissues could be the origin of some cancers. Supportively, polyploid p53-null mammary cells are more prone to tumorigenesis via chromosome missegregation during mitosis than their diploid counterparts[8].

However, other studies have shown that polyploidy may protect from cancer by "buffering" against genotoxic damages and reducing the chance of tumor suppressor loss[9,10]. Importantly, polyploid cells can generate ploidy-reduced progeny via multipolar mitosis[11], which efficiently contributes to organ development as well as regeneration[5,12]. In multipolar mitosis of polyploid cells, three or more daughter cells are produced and in some of these cellular ploidies are reduced by half. In the liver ploidy reduction of binucleated hepatocytes typically produces four daughter cells in a single mitosis[11]. Ploidy reduction co-exists with polyploidization and re-polyploidization in injured livers[5,11]. Given that ploidy reduction could give rise to "unbuffered" daughter cells in only one cell division and additionally induce novel aneuploidy via error-prone multipolar mitosis[13], ploidy reduction may play a key role in the transformation of normal polyploid cells into cancer. Notably, human pan-cancer analysis frequently detected punctuated bursts of chromosome aberrations during tumorigenesis in both polyploid and near-diploid tumors[14]. This implies the existence of a critical mechanism to generate near-diploid tumor cells with extensive chromosome aberrations in a single mitosis, which would reasonably be ploidy reduction via multipolar mitosis. Currently, however, it remains unclear how the dynamic ploidy alterations of normal hepatocytes impact cancer transformation.

Here, we show the involvement of polyploid hepatocytes and their ploidy reduction in liver tumorigenesis. A recently developed method in which cellular ploidy status can be traced is utilized[5]. Genome-wide analysis indicates that proliferation of polyploid hepatocytes give rise to regenerative nodules (RNs) with allelic imbalances (AIs), which are enhanced by ploidy reduction. We also show that polyploid hepatocytes readily initiate tumors while frequently undergoing ploidy reduction. Ploidy reduction is a previously unappreciated mechanism linking normal cells to cancer development.

## Results

**Transplanted polyploid hepatocytes give rise to repopulation nodules with chromosome aberrations via ploidy reduction.** To evaluate the risks of aneuploidy in proliferating polyploid hepatocytes, we analyzed chromosome aberrations at a genome-wide level utilizing F1 hybrids of C57BL/6 and 129S4/SvJae mice heterozygous for *Ubc-CreERT2/Rosa-Confetti*$^{+/-}$ (Fig. 1a and Supplementary Fig. 1a). In mice heterozygous for this multicolor reporter allele, a subset of polyploids are labeled in two colors after Cre recombination and reduction of their ploidy can be traced (Fig. 1a–c)[5]. Using this system, we previously reported that ploidy reduction robustly occurs in multiple distinct kinds of liver injury[5]. Polyploid bicolored hepatocytes were transplanted into *Fah*$^{-/-}$ mice yielding ploidy-reduced monocolored cells and their re-polyploidized progenies as expected (Fig. 1b, c). Monocolored and bicolored cells were then serially transplanted into secondary *Fah*$^{-/-}$ recipients separately to avoid cross-contamination (Supplementary Fig. 1a).

Genome-wide SNP arrays detected many whole-chromosome AIs between C57BL/6 and 129S4/SvJae alleles in clonal repopulation nodules (Fig. 1d and Supplementary Fig. 1b, c). Most AIs reflected gain or loss of whole chromosomes (Supplementary Fig. 1c). Importantly, AIs were more frequent in monocolored RNs than bicolored RNs, suggesting that ploidy reduction promoted chromosome missegregation (Fig. 1e). High-density SNP array confirmed the AIs examined (Supplementary Fig. 1d).

To directly explore chromosomal aberrations in situ, Rosa-mT/EGFP mice useful for tracing chromosome 6 were analyzed after chronic liver injury induced by CCl₄ or thioacetamide (TAA) (Supplementary Fig. 2a). Notably, monocolored regenerative clones were readily found (~0.2% of the sectioned liver areas) in the injured livers of both models (Fig. 1f and Supplementary Fig. 2b–f). Assuming that chromosome 6 is representative, this finding predicts that the frequency of autosome aberrations overall is considerable.

Chromosomal aberrations during liver injury were further explored using SNP-informative *Hgd*$^{+/-}$*Fah*$^{-/-}$ mice, and whole-chromosome loss of the *Hgd* wild-type chromosome 16 in RNs was readily detected (Supplementary Fig. 3). This is consistent with previous studies showing that polyploidy-facilitated aneuploidy of chromosome 16 causes loss of *Hgd* and protects hepatocyte adaptation from Fah deficiency[15,16]. Taken together, these findings indicate that proliferating polyploid hepatocytes in injured livers undergo chromosome aberrations and that ploidy reduction is associated with this process.

**Ploidy reduction is suppressed after continuous proliferation of polyploid hepatocytes.** Given that ploidy reduction results in an enhanced frequency of chromosomal aberrations, repeated cycles of ploidy reduction and polyploidization could be particularly oncogenic. Surprisingly, serially transplanted bicolored polyploids (Supplementary Fig. 1a) produced almost no monocolored progeny in secondary recipients suggesting that the ability to undergo ploidy reduction was lost after multiple cell divisions.

To confirm suppression of ploidy reduction after serial transplantation, bicolored tetraploid hepatocytes were sorted from secondary recipient livers containing both bicolored and monocolored hepatocytes, and transplanted into tertiary recipient mice (Fig. 2a). As expected, ploidy reduction in serially transplanted cells was markedly suppressed, and all (>30) RNs in tertiary recipient mice were bicolored, whereas monocolored cells emerged only as rare oligocellular clusters within bicolored large nodules (Fig. 2b–d). This suppression of ploidy reduction was not due to the long time after polyploidization because aged hepatocytes underwent ploidy reduction like young naïve cells (Supplementary Fig. 4). Loss of ploidy reduction in serially transplanted polyploid hepatocytes was also confirmed in the context of CCl₄-liver injury (Supplementary Fig. 5). Therefore, ploidy reduction in polyploid hepatocytes is suppressed after continuous proliferation during serial transplantation.

Next, clues to the mechanism(s) for ploidy reduction were sought. We previously hypothesized that binucleation in polyploids is important for proper chromosome segregation during ploidy reduction[5]. Notably, nuclear numbers in polyploid cells were quite different between naive and serially transplanted tetraploid cells. Serially transplanted tetraploids were mostly mononucleated (Fig. 2e). Moreover, centrosome amplification, which is induced during polyploidization and closely related to occurrence of multipolar mitosis, was mostly reversed in serially transplanted polyploids (Fig. 2f–h). These findings suggest that after continuous proliferation polyploid hepatocytes become mostly mononucleated and monocentrosomal, leading to suppression of ploidy reduction.

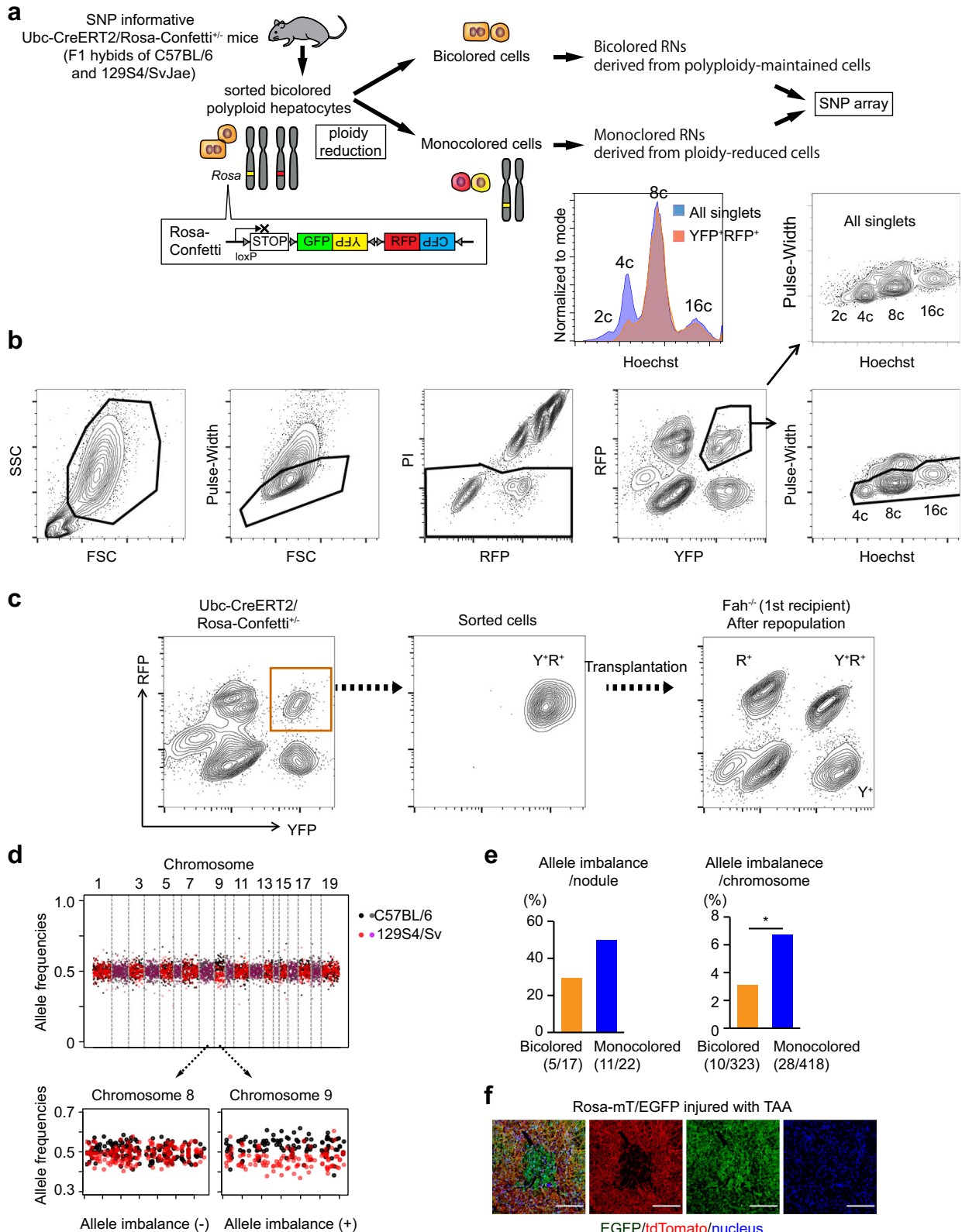

**Fig. 1 Regenerative nodules in injured livers harbor chromosome aberrations via ploidy reduction. a** Experimental overview. Detailed scheme is shown in Supplementary Fig. 1a. **b** Sorting strategy for YFP+RFP+ polyploid hepatocytes (lower panel) and comparison of ploidy with all hepatocytes (upper panel). Hepatocytes were sorted based on both their DNA content (Hoechst) and fluorescent reporter expression. Doublets were strictly excluded by narrow gating with the pulse-width parameter. **c** Representative FACS data at initial and serial sorting. **d** Representative plots of strain-specific allele frequencies in RNs. Only informative SNPs that are different between C57BL/6 and 129S4Sv are plotted. Note that consistent shift of allele frequencies from 0.5 indicates AIs. **e** Frequencies of AIs in RNs. Values in parentheses indicate numbers of RNs or autosomes with AIs among those analyzed. *$p = 0.028$ (chi square test without adjustments for multiple comparisons). **f** Microscopic images of a RN with loss of heterozygosity. Mice were injured with TAA for about 10 weeks. Scale bars, 100 μm. FSC forward scatter, SSC side scatter.

**Polyploid hepatocytes readily form tumors in several models of liver cancer.** Next, the potential of polyploid hepatocytes as cancer cells of origin was directly examined using different hepatocarcinogenesis models. Heterozygous Rosa-RGBow multi-reporter mice were hydrodynamically injected with a Cre-encoding plasmid as well as plasmids encoding sleeping beauty transposase and oncogenic transposons (Fig. 3a)[17]. Hepatocytes transduced hydrodynamically by plasmids were predominantly (96.6 ± 1.9%) polyploid presumably due to biased distribution of polyploid hepatocytes around pericentral areas (Supplementary Fig. 6a, b)[18,19]. Delivery of constitutively active Akt and Myc or Yap oncogenes (Akt/Myc or Akt/Yap) induced multiple liver tumors (Fig. 3b). Tumors induced by Akt/Myc were histologically consistent with hepatocellular carcinoma (HCC), whereas Akt/Yap-induced tumors contained cholangiocellular carcinoma as well as HCC components (Fig. 3c) as reported[17]. Importantly, 37.9% (Akt/Myc) and 29.8% (Akt/Yap) of tumors were bicolored regardless of the histological tumor type, indicating robust tumorigenic potential in polyploid hepatocytes (Fig. 3d). Transformation of bicolored polyploids into both HCC and cholangiocellular carcinoma was also confirmed by hydrodynamic injection into pre-labeled Rosa-RGBow$^{+/-}$ mice (Fig. 3e). Notably, the frequency of bicolored tumors among labeled tumors was significantly lower ($p < 0.05$ for Akt/Myc and $p < 0.01$ for Akt/Yap) than at baseline (47.7 ± 2.0%) (Fig. 3d). One explanation for the decreased frequency of bicolored tumors in this polyploid-derived tumorigenesis model is that some polyploids underwent ploidy reduction during carcinogenesis.

Next, heterozygous Rosa-RGBow mice labeled with AAV8-Ttr-Cre[5] were subjected to three kinds of tumor-prone chronic liver injuries including TAA injury[20], nonalcoholic steatohepatitis (NASH)[21], and Fah deficiency[22]. Administration of AAV8-Ttr-Cre ($6 \times 10^{10}$ vector genomes (vg)) efficiently labeled hepatocytes and reflected their ploidy distribution (Supplementary Fig. 6c, d). After several months, all models developed tumors including bicolored HCCs, confirming that polyploids give rise to tumors even without forced expression of oncogenes (Fig. 3f–i). The frequencies of bicolored tumors were lower than the labeling baselines in all injuries and were similar to or lower than chronically injured livers that underwent ploidy reduction (Fig. 3h)[5]. This implies ploidy reduction during carcinogenesis induced by chronic injury. Taken together, tracing of bicolored polyploids in various kinds of liver cancer models indicated that polyploid hepatocytes readily serve as a source of tumors, and implied that ploidy reduction can occur during polyploid-derived cancer development.

**Ploidy reduction frequently occurs during polyploid-derived carcinogenesis.** Decreased frequencies of bicolored tumors compared to those of bicolored cells at baseline (Fig. 3d, h) can be explained not only by ploidy reduction but also by higher tumorigenic potential of diploids[9]. If the theoretical assumption is made that no ploidy reduction occurs during carcinogenesis, diploid cells were indeed calculated to have a much higher (>>10 times) tumorigenic potential than polyploids. Supplementary Fig. 6e, f shows the theoretical tumor potential of diploid cells with and without ploidy reversal.

Therefore, to directly measure the tumorigenic potential of diploids and polyploids, a competitive oncogenesis assay to test diploid, tetraploid, and octaploid hepatocytes was performed. Wild-type, tdTomato$^{+/+}$, and GFP$^{+/+}$ hepatocytes sorted by ploidy were subjected to in vitro targeting of the $p53$ and $Pten$ tumor suppressor genes with CRISPR/Cas9 in a defined mixture, and subsequently transplanted into Fah$^{-/-}$ mice (Supplementary Fig. 7a, b). After a few months multiple tumors developed, and their origins were

examined by reporter expression (Supplementary Fig. 7c). In this competitive tumor formation assay, the cancer initiation potentials of tetraploid and octaploid cells were marginally reduced compared to diploids, but not nearly as low as expected without ploidy reduction (Fig. 4a and Supplementary Fig. 6f). In combination with the decreased frequencies of bicolored tumors (Fig. 3d, h), these findings are indicative of ploidy reduction during polyploid-derived carcinogenesis.

To further investigate polyploid-derived tumorigenesis in detail, bicolored polyploid hepatocytes were sorted from heterozygous Rosa-Confetti$^{+/-}$ or $p53^{+/-}$Rosa-Confetti$^{+/-}$ mice, and transplanted into Fah$^{-/-}$ recipients after mutagenesis in $p53$, $Pten$, or $Cdkn2a$ genes (Fig. 4b). Interestingly, the majority of tumors derived from bicolored polyploids lost either one or both reporters, suggesting frequent ploidy reduction during polyploid-derived tumorigenesis in this system (Fig. 4c, d). Consistent occurrence of ploidy reduction irrespective of $p53$ knockout frequencies suggested that ploidy reduction did not result from loss of $p53$ (Fig. 4e). Although some tumors exhibited segmental loss of reporter expression, the loss of reporters was mostly observed in the entire tumor, indicating that ploidy reduction occurred in the early phase of tumorigenesis (Fig. 4f, g). Notably, among 30 tumors with intratumoral ploidy reduction analyzed, 2 tumors were accompanied by monocolored cholangiocellular transdifferentiation, which was observed at the border between a bicolored tumor portion and a monocolored one in both tumors (Fig. 4h and Supplementary Fig. 8). Close anatomical correlation between transdifferentiation and ploidy reduction in these tumors implies that transdifferentiation emerged from bicolored hepatocellular tumor cells via ploidy reduction, and supports involvement of ploidy reduction in tumor evolution.

In marked contrast to tumorigenesis from naive hepatocytes, serially transplanted polyploid hepatocytes, whose ploidy reduction is suppressed, rarely underwent ploidy reduction even during tumorigenesis with $p53$ knockout (Supplementary Fig. 9). Polyploids that have proliferated extensively resist ploidy reduction even under oncogenic conditions. Taken together, these data indicate that proliferative polyploid hepatocytes form tumors with only slightly lower frequency than diploid counterparts and that ploidy reduction frequently occurs especially in the early phase of polyploid-derived tumorigenesis.

**Ploidy reduction promotes polyploid-derived tumor development.** Taking advantage of the fact that serially transplanted hepatocytes are resistant to ploidy reduction, we performed competitive oncogenesis assay to determine whether ploidy reduction promotes tumorigenesis. Sorted serially transplanted and naive polyploid hepatocytes were subjected to $p53$ and $Pten$ mutagenesis in a mixture, and their tumor initiating frequencies were compared after transplantation (Fig. 5a). tdTomato$^{+/+}$ polyploids which will remain labeled even after ploidy reduction as well as Rosa-RGBow$^{+/-}$ polyploids whose ploidy reduction is detectable were used as naive hepatocytes. In both tetraploids and octaploids, naive hepatocytes had significantly higher tumorigenic potential than serially transplanted polyploids (Fig. 5b). Taking it into account that serially transplanted hepatocytes are capable of proliferation for more than seven rounds of liver repopulation[23] and are reported to maintain similar proliferative capacities compared to naïve cells[24], this finding is consistent with the hypothesis that ploidy reduction promotes polyploid-derived tumorigenesis.

Given the nature of ploidy reduction, it might enhance carcinogenesis via inducing oncogenic aneuploidies and/or loss of intact alleles of tumor suppressor genes. SNP array analysis of tumors derived from $p53^{+/-}Confetti^{+/-}$ polyploid hepatocytes

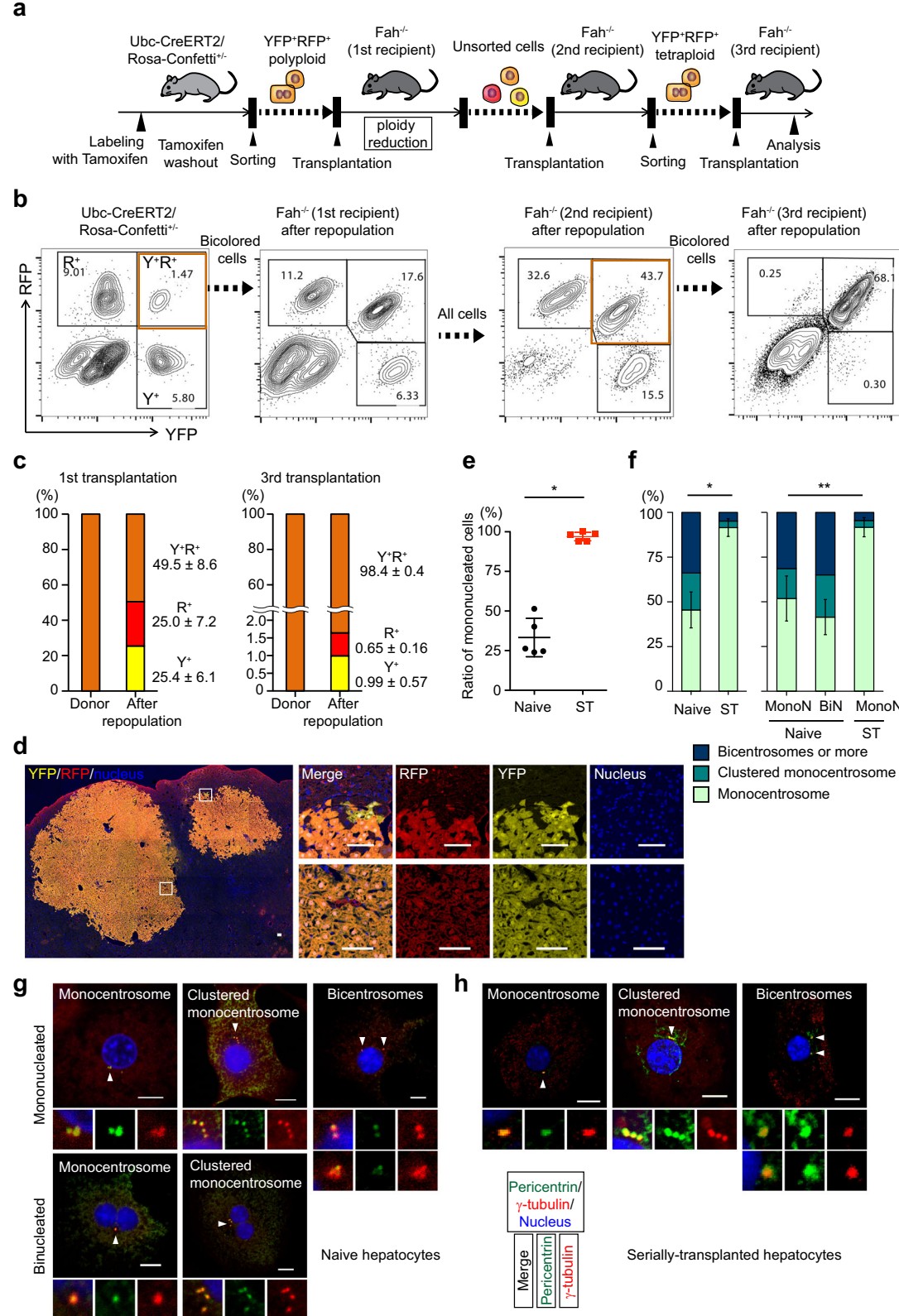

detected AIs in both bicolored and ploidy-reduced (reporter-lost) tumors (Supplementary Fig. 10a). One of the chromosomes that frequently had AIs was chromosome 19 encoding *Pten*. Chromosome 19 AIs were enriched in reporter-lost tumors compared to bicolored tumors (Supplementary Fig. 10b). Interestingly, although the knockout efficiency of the *Pten* gene in the input cells was only about 60%, the insertion/deletion (indel) frequencies in *Pten* was near 100% in all tumors examined. Furthermore, ploidy-reduced tumors had less variability in the kinds of indels present than bicolored tumors (Supplementary Fig. 10c–e). These findings suggest that ploidy reduction promoted tumor development by inducing loss of *Pten*-intact

**Fig. 2 Suppression of ploidy reduction in serially transplanted hepatocytes. a** Experimental scheme of serial transplantation. **b** Representative FACS plots of donor and serial recipient livers. The percentages of each fraction are shown. Note that proportion among colored cells is consistent before and after second transplantation, indicating similar proliferative capacity among colored cells. **c** Frequencies of colored cells before and after first and third transplantation. Values are means ± SD ($n = 5$ in the first transplantation, and $n = 3$ in third transplantation). **d** Microscopic images of tertiary recipient liver. Scale bars, 100 μm. **e** Ratio of mononucleated cells. Error bars indicate mean ± SD ($n = 5$ biologically independent mice in each group). *$p = 3.0 \times 10^{-6}$ (two-sided unpaired $t$-tests without adjustments for multiple comparisons). **f** Centrosome numbers in naive and serially transplanted tetraploid hepatocytes. Cells sorted from biologically independent mice were analyzed. Error bars indicate mean ± SD of monocentrosomal frequencies ($n = 3$ in naive hepatocytes, and $n = 5$ in serially transplanted hepatocytes). *$p = 0.00014$, **$p = 0.018$ (two-sided unpaired $t$-tests without adjustments for multiple comparisons). Representative microscopic images of centrosome staining of naïve hepatocytes (**g**) and serially transplanted hepatocytes (**h**). Scale bars, 10 μm. In (**e–h**), 494 serially transplanted hepatocytes (five mice) and 350 naive hepatocytes (four mice) were analyzed. Naive hepatocytes were collected from 12-week-old mice. Serially transplanted hepatocytes were collected from secondary or tertiary recipient mice by sorting bicolored polyploid cells. $Y^+$ YFP$^+$, $R^+$ RFP$^+$, ST serially transplanted, MonoN mononucleate, BiN binucleate. Source data are provided as a Source Data file.

and enrichment of *Pten*-deficient chromosomes 19, via AIs and/or ploidy reduction in itself. Taken together, ploidy reduction was shown to promote cancer development derived from polyploid hepatocytes.

## Discussion

Several recent studies have examined the role of ploidy in cancer formation particularly in the liver, and concluded that polyploidy protects hepatocytes from tumorigenesis[9,10,25]. These studies indirectly estimated the effects of ploidy status on tumorigenesis by comparing livers whose ploidy status was genetically or metabolically manipulated[9,10,25]. In contrast, we here directly compared the oncogenic potential of hepatocytes with differing ploidy by tracking in the same liver. Consistent with these previous reports[9,10,25], polypoid oncogenicity was lower than that of diploids in cancers induced by tumor suppressor loss, supporting the "buffering" effect of polyploidy (Fig. 4a). Importantly, however, this difference was less than twofold and tumorigenesis from polyploid hepatocytes was quite robust if ploidy reduction is considered. Bicolored tumors in livers with unmanipulated mixed-ploidy also provided the first direct evidence that polyploids generate tumors.

We showed that ploidy reduction frequently occurred during an early phase of tumorigenesis, and enhanced polyploid-derived cancer development. One possible mechanism of carcinogenesis mediated by ploidy reduction is enhanced chromosomal instability (CIN), the process generating aneuploidy. Others have previously shown that mitosis of polyploid cells leads to CIN[3,7]. However, the proposed mechanism involves loss of one or a few chromosomes by missegregation during a (pseudo-)bipolar mitosis[3,13]. This mechanism is very different from the CIN that occurs during ploidy reduction divisions of hepatocytes[11]. Ploidy reduction in hepatocytes is a process distinct from (pseudo-) bipolar mitosis in that ploidy is reduced by half in a single cycle of multipolar cell division. Here, we showed that ploidy reduction (multipolar mitosis) not only results in more CIN than bipolar mitosis in polyploids, but also promotes tumor initiation by reducing the chromosome number by half in a single mitosis.

Previous studies in which hepatocyte aneuploidies in healthy or cirrhotic livers were analyzed by sequencing of single cells or microdissected tissues revealed that aneuploidy was rare in normal hepatocytes but occasionally observed in RNs in cirrhosis[26,27]. These findings suggest that hepatocytes acquire aneuploidy during regenerative proliferations. Consistently, we here demonstrate whole-chromosome imbalances of paternal and maternal chromosomes in polyploid-derived RNs, which were especially augmented after ploidy reduction. While our recent study showed largely normal chromosome segregation during ploidy reduction based on tracing of a single autosome, genome-wide analysis in the present study refined our model with an estimated chromosome aberration rate during ploidy reduction of

~4.5% per chromosome (Fig. 1c). Moreover, we also demonstrated that those ploidy-reduced cells with chromosome aberrations can clonally proliferate in vivo. These findings suggest that CIN in proliferating polyploid hepatocytes is enhanced by ploidy reduction, which could promote cancer development. Ploidy reduction accompanied with CIN during carcinogenesis is also supported by recent analysis of human pan-cancer genomes[14].

Ploidy reduction could also promote carcinogenesis by reversing the tumor protective role of the redundant polyploid genome. Polyploid cells are thought to be well protected from transformation by loss of heterozygosity (LOH). However, our data (Fig. 4) clearly show that even octaploid hepatocytes are vulnerable to LOH oncogenesis. This apparent paradox is explained by the fact that polyploid hepatocytes rapidly and frequently become diploid during regeneration. Thus, ploidy reduction puts polyploids on the same level of LOH vulnerability as diploids. Comparison of indel variants (Supplementary Fig. 10c–e) supported that ploidy reduction induced loss of intact alleles and condensed oncogenic alleles. Such an oncogenic mutant allele imbalance is widely observed in human cancers[28]. Multipolar mitosis that leads to CIN as well as loss of tumor suppressive alleles in a single event would critically trigger cancer initiation in polyploid cells with tumor suppressor mutations.

Both of the proposed mechanisms by which ploidy reduction promotes carcinogenesis are based on enhancement of tumor suppressor loss. Consistently, in tumorigenesis models induced by tumor suppressor loss, the majority of tumors (around 80%) underwent ploidy reduction during the early steps of carcinogenesis (Fig. 4d). On the other hand, ploidy reduction was less frequent (<37%) in oncogene-induced cancer models (Fig. 3d). In the spontaneous carcinogenesis models mediated by chronic liver injuries, estimated ploidy reduction frequencies were between the LOH and oncogene-activation models (<56–78%, Fig. 3h). These findings indicate that ploidy reduction disables the protective effect of polyploidy against LOH, but is not essential in injury-induced tumors that could involve various tumorigenic mechanisms other than tumor suppressor loss. Further studies would be required to determine whether ploidy reduction affects oncogene-induced carcinogenesis, and to what extent it promotes overall tumorigenesis under clinically relevant conditions.

Notably, ploidy reduction was suppressed in polyploids after prolonged proliferation. Polyploidized hepatocytes generated via incomplete cytokinesis are binucleated and bicentrosomal[29], and amplified centrosomes lead to error-prone mitosis including multipolar mitosis[30]. Our results suggested that mononucleation and centrosome clustering during proliferation[31] led to suppression of ploidy reduction. The decreased tumorigenesis observed in serially transplanted hepatocytes compared to binucleated naive polyploids (Fig. 5) is consistent with the hypothesis that ploidy reduction is important in LOH-induced carcinogenesis of polyploids. However, serially transplanted hepatocytes may differ from their naive counterparts in not only their capacity for

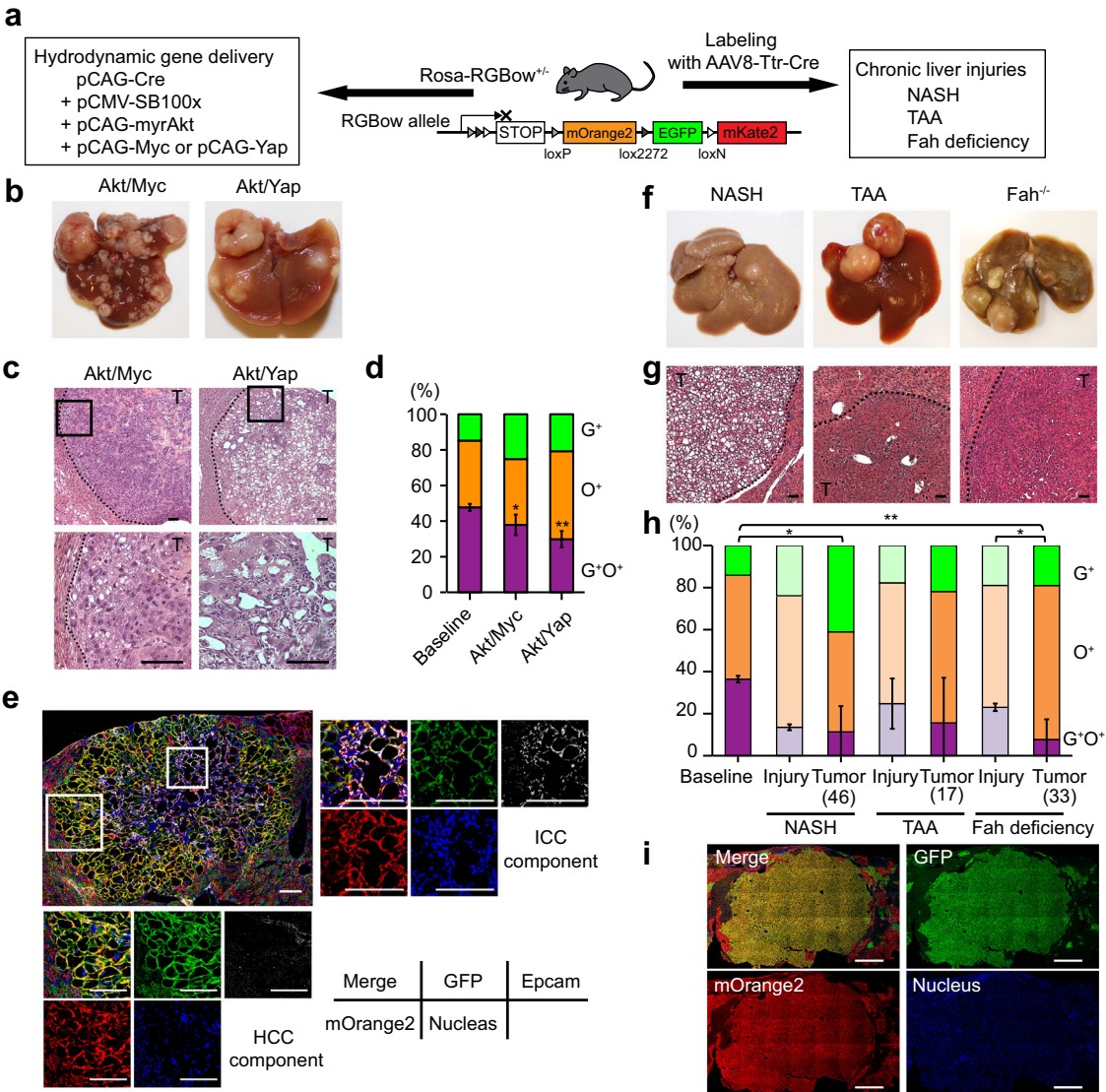

**Fig. 3 Tracing of polyploid-derived hepatocarcinogenesis in Rosa-RGBow[+/−] mice. a** Experimental scheme of tumor induction in *Rosa-RGBow[+/−]* mice. Representative macroscopic (**b**) and HE staining (**c**) images of tumors induced by hydrodynamic injection. High-magnification views are shown in lower panels. Scale bars, 50 μm. **d** Frequencies of bicolored and monocolored tumors induced by hydrodynamic injection. Biologically independent mice were analyzed. Error bars indicate mean ± SD ($n = 5$ in baseline and Akt/Myc, and $n = 3$ in Akt/Yap). *$p = 0.012$, **$p = 0.006$ (two-sided unpaired *t*-tests without adjustments for multiple comparisons). **e** Bicolored tumor consistent with both HCC and ICC component in a pre-labeled *Rosa-RGBow[+/−]* mouse. High-magnification images show Epcam-positive ICC component and Epcam-negative HCC component. Note that RGBow reporters exhibit membranous expressions. Scale bars, 100 μm. Representative macroscopic (**f**) and HE staining (**g**) images of chronically injured livers developing tumors. Scale bars, 50 μm. **h** Frequencies of bicolored and monocolored tumors in chronically injured livers. Frequencies of labeled cells before initiation of liver injuries are also shown as a baseline. Biologically independent mice were analyzed. Error bars indicate mean ± SD ($n = 3$ in baseline, Fah deficiency-induced and NASH-induced injury, $n = 4$ in TAA-induced injury, $n = 7$ in Fah deficiency-induced carcinogenesis, and $n = 8$ in NASH- and TAA-induced carcinogenesis). Frequencies in TAA injury and Fah-deficient models are previously shown[5]. Values in parentheses indicate total numbers of tumors analyzed. *p* values were calculated by two-sided unpaired *t*-tests without adjustments for multiple comparisons. *$p < 0.05$ ($p = 0.011$, baseline vs. NASH-induced tumor; $p = 0.039$, Fah-deficient injury vs. Fah deficiency-induced tumor) **$p < 0.01$ ($p = 0.0017$, baseline vs. Fah deficiency-induced tumor). **i** Representative microscopic images of a bicolored tumor in chronically NASH injured liver. Scale bars, 1 mm. Note that mKate2[+] cells were not analyzed because they were quite rare (<1% of labeled cells)[5]. T tumor, G[+] EGFP[+], O[+] mOrange2[+]. Source data are provided as a Source Data file.

ploidy reduction, but also other properties that affect oncogenesis. Therefore, future studies will be needed to further confirm the "ploidy reduction hypothesis."

Overall, we conclude that polyploid hepatocytes are not fully protected from oncogenesis and that ploidy reduction can promote polyploid-derived cancer initiation. In addition to the argument that multipolar mitosis leading to ploidy reduction promotes carcinogenesis via inducing CIN, fluctuation of ploidy (polyploidization and subsequent ploidy reduction) may be

involved in cancer evolution, like vertebrate evolution[32]. Given that polyploidy is sometimes observed even in diploid tissues such as intestinal epithelia with chronic damages[33], ploidy reduction could trigger cancer initiation in various tissues.

## Methods
**Mice and Cre recombination.** *Ubc-CreERT2*[34], *Rosa-Confetti*[35], *Rosa-RGBow*[36], *Rosa-mTomato/mGFP* (Rosa-mTmG)[37], *Rosa-Cas9EGFP*[38], and C57BL/6 wild-type mice were obtained from The Jackson Laboratory and maintained on the

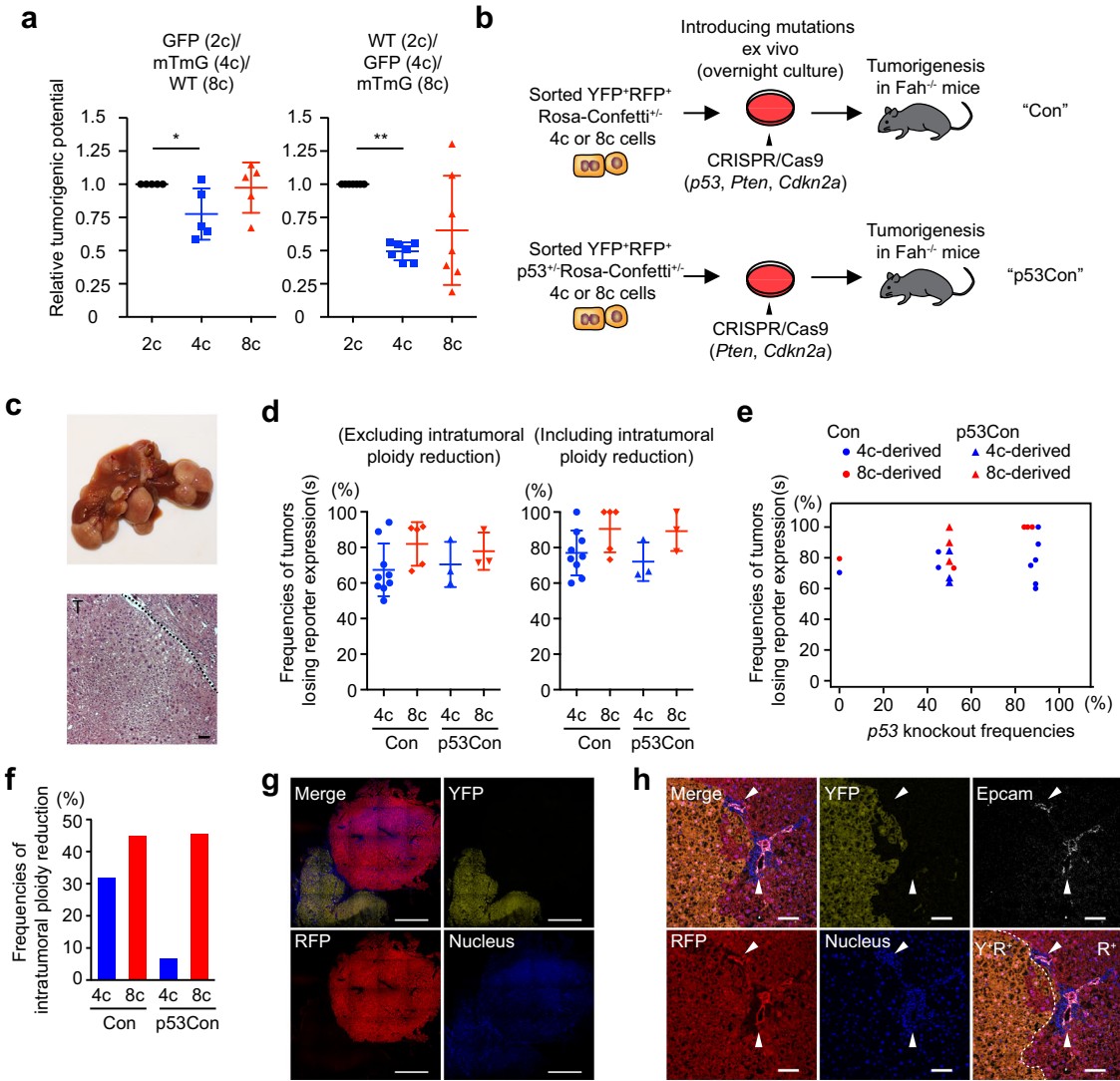

**Fig. 4 Risk evaluation and frequent ploidy reduction in polyploid-derived tumorigenesis. a** Comparison of tumorigenic potential between diploid, tetraploid, and octaploid hepatocytes. Relative tumorigenic potential compared to diploids is shown. Biologically independent recipient mice were analyzed, and error bars indicate mean ± SD ($n = 5$ and 7 in GFP(2c)/mTmG(4c)/WT(8c) and WT(2c)/GFP(4c)/mTmG(8c), respectively). For additional details see Supplementary Fig. 7. *$p = 0.03$, **$p = 3.1 \times 10^{-6}$ (two-sided paired $t$-tests without adjustments for multiple comparisons). **b** Experimental scheme to trace tumorigenesis derived from polyploid hepatocytes. **c** Representative macroscopic and HE staining images of polyploid-derived tumors. T tumor. Scale bar, 50 μm. **d, e** Frequencies of tumors that lost reporter expression(s) among YFP⁺RFP⁺ polyploid-derived tumors. Error bars indicate mean ± SD. Correlation with $p53$ knockout frequencies in origin cells is shown in (**e**). Tumors were analyzed about 4 months after transplantation, and 17.8 tumors were analyzed in each mouse on average. **f** Frequencies of intratumoral ploidy reduction among multicolored tumors. In each group, 11–47 tumors were analyzed. **g** Microscopic images of monocolored tumors derived from YFP⁺RFP⁺ polyploids. Scale bars, 1 mm. **h** Representative microscopic images of intratumoral ploidy reduction accompanied by cholangiocellular transdifferentiation. Transdifferentiation into Epcam-positive cholangiocellular cells were indicated by arrow heads. Scale bars, 100 μm. Con YFP⁺RFP⁺ *Rosa-Confetti*⁺ᐟ⁻ derived, p53Con YFP⁺RFP⁺ *p53*⁺ᐟ⁻*Rosa-Confetti*⁺ᐟ⁻ derived, Y⁺ YFP⁺, R⁺ RFP⁺. Source data are provided as a Source Data file.

C57BL/6 background. *Trp53* mutant mice were obtained from Dr. Allan Bradley (Department of Medicine, University of Cambridge) and maintained on the 129S4/SvJae background[39]. 129S4/SvJae wild-type mice and Rosa-mTmG mice on the 129S4/SvJae background were obtained from Dr. Philippe M. Soriano (Icahn School of Medicine at Mount Sinai). GFP transgenic mice[40] were obtained from Dr. Masaru Okabe (Osaka University) and backcrossed with 129S4/SvJae more than ten times. *Fah*⁻ᐟ⁻ mice on either the C57BL/6 or the 129S4/SvJae background[41] were maintained on 8 mg/L of 2-(2-nitro-4-trifluoromethylbenzoyl)-1,3-cyclo-hexanedione (NTBC; Yecuris Corp., Portland, OR) in drinking water until transplantation. In the competitive oncogenesis assay to test diploid, tetraploid, and octaploid hepatocytes, 129S4/SvJae wild-type mice, *tdTomato*⁺ᐟ⁺ and *GFP*⁺ᐟ⁺ hepatocytes on the 129S4/SvJae background were sorted by ploidy and transplanted into *Fah*⁻ᐟ⁻ mice on the same 129S4/SvJae background. *Ubc-CreERT2/Rosa-Confetti*⁺ᐟ⁻ mice were given tamoxifen (Sigma-Aldrich, St Louis, MO) dissolved in corn oil (Sigma-Aldrich, 20 mg/ml) at a dose of 150 mg/kg body weight three to five times to induce Cre recombination[5] and were processed for transplantation

after a tamoxifen washout period of at least 3 weeks. In cancer models by hydrodynamic tail vein injection, Cre-encoding plasmid was injected into mice along with oncogene-encoding plasmids unless otherwise specified. Cre recombination in the other mice was induced by retro-orbital administration of AAV8-Ttr-Cre[42] into 8-week-old or older mice at a dose of $6 \times 10^{10}$ vg per mouse. Mice were housed in the Department of Comparative Medicine at Oregon Health and Science University. Mice were maintained under temperature- and humidity-controlled conditions and were exposed to on a 12-h light–dark cycle. The Oregon Health & Science University Institutional Animal Care and Use Committee (Portland, OR) approved all animal experiments described.

**Plasmid construction and hydrodynamic tail vein injection.** pCMV(CAT)T7-SB100 encoding SB100X transposase was obtained from Addgene (Addgene plasmid # 34879)[43]. pT3-CAG-myrAKT and pT3-CAG-YAP were constructed by modifying pT3-myr-AKT-HA (Addgene plasmid # 31789)[44] and pT-3EF1a-Yap

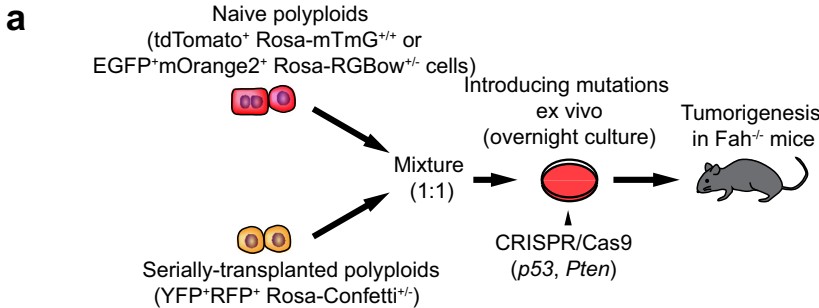

**Fig. 5 Enhanced polyploid-derived tumorigenesis by ploidy reduction. a** Experimental scheme of a competitive experiment using naive and serially transplanted polyploid hepatocytes. Serially transplanted YFP⁺RFP⁺ polyploid cells heterozygous for Rosa-Confetti were sorted from secondary or tertiary recipient Fah⁻/⁻ mice. **b** Ratios of tumor numbers derived from naive and serially transplanted hepatocytes. Tumors were analyzed about 4 months after transplantation. Data are shown as relative values compared to serially transplanted hepatocytes. Values in parentheses indicate bicolored and ploidy-reduced (either monocolored or non-colored) tumors. *p* values were calculated by one-sided paired *t*-tests without adjustments for multiple comparisons. Tom⁺ tdTomato⁺, G⁺ EGFP⁺, O⁺ mOrange2⁺, Y⁺ EYFP⁺, R⁺ RFP⁺.

S127A (Addgene plasmid # 46049)[45], respectively. Briefly, a loxP site and EF-1α promoter were removed from the original plasmids, and the chimeric CMV early enhancer/chicken β actin (CAG) promoter was inserted to create pT3-CAG-myrAKT and pT3-CAG-YAP. pT3-CAG-Myc was constructed by substituting the myristoylated AKT gene cassette in pT3-CAG-myrAKT with mouse *Myc* gene. pCAG-Cre was created by substituting CreEGFP fusion cassette in CAG-GFP/Cre (Addgene plasmid # 49054)[46] with Cre recombinase gene. Plasmids were injected into the tail vein of heterozygous Rosa-RGBow mice of about 20 grams of body weight within 5–7 s in a volume of saline equivalent to 10% of the body weight. The amount of injected DNA was 15 µg for pT3-CAG-myrAKT, 15 µg for pT3-CAG-YAP or pT3-CAG-Myc, 7.5 µg for pCMV(CAT)T7-SB100, and 7.5 µg for pCAG-Cre. In injection into pre-labeled Rosa-RGBow⁺/⁻ mice, Rosa-RGBow⁺/⁻ mice received hydrodynamic injection of pCMV-SB100x, pCAG-myrAkt, and pCAG-Yap at 2 weeks after administration of AAV8-Ttr-Cre.

**Liver injury inductions**. TAA (Sigma-Aldrich) was diluted in drinking water (0.03%, w/v) and continuously administered to mice for about 11 months[20]. For induction of NASH-related HCCs, mice were fed high-fat, high-fructose, and high-cholesterol Western diet (Taklad diet TD.120528, Envigo, Madison, WI) and a high sugar water containing d-fructose (23.1 g/L) and d-glucose (18.9 g/L), and administered with low-dose CCl4 (Sigma-Aldrich, 0.2 µl/g body weight, once per week) for 6–8 months[21]. *Fah* knockout mice were put on cycling withdrawal of NTBC (2–3 weeks off and 3–5 days on) for 6–8 months[41]. To induce CCl4-mediated liver

injury, CCl4 (Sigma-Aldrich) was diluted in corn oil (1:4, v/v) and intraperitoneally injected twice a week at a dose of 5 ml/kg body weight[47]. To analyze ploidy reduction during CCl4-induced liver injury, sorted cells were transplanted into strain-matched wild-type mice, and mice were administered with CCl4 for 6 weeks from 2 weeks after transplantation.

**Flow cytometry and fluorescence-activated cell sorting (FACS) of primary hepatocytes**. Primary hepatocytes were isolated by a two-step collagenase perfusion method[48], and collected by sequential centrifugation at $50 \times g \times 2$ min. Collected cells were suspended in Dulbecco's modified Eagle's medium with 10% fetal bovine serum and 10 mM HEPES. Cells were incubated with 15 mg/ml Hoechst 33342 (Thermo-Fisher Scientific, Waltham, MA) and 5 mM reserpine (ThermoFisher Scientific) for 30 min at 37 °C to detect ploidy, and stained with propidium iodide (Sigma-Aldrich) to exclude dead cells. FACS was performed with a Cytopeia inFluxV-GS (Becton Dickenson, Franklin Lakes, NJ) at a flow rate of about 1000 cells/s. Flow cytometric analysis was performed with a BDLSRFortessa (Becton Dickenson), BD FAC-Symphony (Becton Dickenson), or Cytoflex S (Beckman Coulter Life Sciences, Indianapolis, IN). Data were processed using FlowJo software (TreeStar).

**Transplantation of sorted cells with and without genome editing**. When genome editing was not performed, sorted cells were resuspended in phosphate-buffered saline and transplanted into background-matched Fah⁻/⁻ mice by intra-splenic injection. To perform genome editing, sorted cells were plated on collagen-

coated dishes, and incubated for 4 h until transfection. Ribonucleoprotein complex was transfected into plated cells using Cas9 protein (Integrated DNA Technologies, Coralville, IA), modified synthetic gRNAs (Synthego, Menlo Park, CA), and Lipofectamine CRISPRMAX Cas9 Transfection Reagent (ThermoFisher Scientific) according to the manufacturers' instructions. Sorted bicolored Rosa-Confetti$^{+/-}$ polyploid hepatocytes were transfected with Cas9/gRNA complex targeting *p53*, *Pten*, and/or *Cdkn2a*, and sorted p53$^{+/-}$ Rosa-Confetti$^{+/-}$ hepatocytes were subjected to genome editing targeting *Pten* and *Cdkn2a*. Sequences of gRNAs are as follows: *p53*, 5′-CCTCGAGCTCCCTCTGAGCC-3′ or 5′-CTGAGCCAGGAGACA TTTTC-3′; *Pten*, 5′-AGATCGTTAGCAGAAACAAA-3′; *Cdkn2a*, 5′-GTGCGATA TTTGCGTTCCGC-3′. Transfected cells were incubated overnight at 37 °C, and collected by trypsinization. Cells resuspended in phosphate-buffered saline were transplanted into *Fah*$^{-/-}$ mice as described above. NTBC was withdrawn from *Fah*$^{-/-}$ recipient mice following transplantation and cycled until the liver was repopulated[48].

**Evaluation of nuclear and centrosome numbers in primary hepatocytes**. Sorted primary hepatocytes were plated on collagen-coated chamber slides, and incubated overnight at 37 °C. Then, cells were fixed with methanol and immunostained with rabbit anti-Pericentrin (1:1000, abcam, ab4448), Alexa Fluor 488-conjugated anti-rabbit (1:200, Jackson ImmunoResearch), and Alexa Fluor 647-conjugated anti-gamma Tubulin (1:1000, abcam, ab191114). To eliminate false positive signals from nonspecific immunostaining, centrosomes were identified as structures positive for both Pericentrin and gamma Tubulin. Nuclei were stained with Hoechst, and the nuclear number in each cell was evaluated on microscopy.

**Histology, immunofluorescence, and imaging of liver tissues**. Liver tissue was fixed in 4% paraformaldehyde (Sigma-Aldrich), and embedded in paraffin or optimum cutting temperature compound (Leica Instruments, Nussloch, Germany). Paraffin-embedded tissues were subjected to hematoxylin and eosin. Frozen sections were immunostained with primary and secondary antibodies including rat anti-Epcam (1:100, Becton Dickenson, 552370), rabbit anti-Fah (1:100)[48], Alexa Fluor 647-conjugated anti-rat (1:200, Jackson ImmunoResearch), and Alexa Fluor 647-conjugated anti-rabbit (1:200, Jackson ImmunoResearch), and counterstained with Hoechst 33342. 3D imaging analysis to examine ploidy reduction during CCl4 injury was performed by tissue clearing with N-methylacetamide (ThermoFisher Scientific), Omnipaque 350 (GE Health- care), Triton X-100 (0.1% v/v, ThermoFisher Scientific), and 1-thioglycerol (0.5% v/v, Sigma-Aldrich)[5]. Fluorescent images were obtained with a Zeiss LSM 780 and LSM 700 confocal microscope (Carl Zeiss AG, Jena, Germany), and analyzed using Zeiss Zen software (Carl Zeiss AG).

**SNP array analysis**. RNs and liver tumors were isolated using stereoscopic microscope, and DNA was isolated using the MasterPure Complete DNA/RNA Purification kit (Epicentre Biotechnologies, Madison, WI). SNP array analysis was performed by Neogen Genomics (Lincoln, NE), an Illumina Infinium certified provider via the standard Infinium protocols provided by Illumina (San Diego, CA). MiniMUGA array with 10,171 markers was performed for routine SNP analysis. GigaMUGA[49] with 143,259 probes was used for high-density SNP array analysis. Obtained signals were processed by GenomeStudio software (Illumina) to calculate b allele frequencies (BAFs) and log r ratios (LRRs).

**Detection of allele imbalances**. To define informative SNPs that are different between 129S4/SvJae and C57BL/6, consensus genotypes of these two strains were obtained from the Mutant Mouse Resource and Research Center at University of North Carolina (https://www.med.unc.edu/mmrrc/genotypes/resources). We first filtered out SNPs with a GenCall score (GC) < 0.15, as recommended by Illumina. To exclude strain-specific variant SNPs that are different from consensus genotypes, SNPs whose mean BAF values among 39 *Ubc-CreERT2/Rosa-Confetti*$^{+/-}$-derived RNs or 39 *p53*$^{+/-}$*Rosa-Confetti*$^{+/-}$-derived tumors were between 0.4 and 0.6 were further selected. Consequently, 2248 and 2296 SNPs on autosomes are defined as informative SNPs among 10,171 markers in the MiniMUGA array, in RN and tumor analysis, respectively, and 129S4/SvJae and C57BL/6 allele frequencies in these informative SNPs were calculated as BAFs or "1-BAFs" based on their genotypes. As both nodule/tumor cells and contaminating *Fah*$^{-/-}$ recipient cells are derived from F1 mice of two strains, both 129S4/SvJae and C57BL/6 allele frequencies should be randomly distributed around 0.5 regardless of contaminating cells. However, if clonal nodule/tumor cells have allele imbalances, C57BL/6 allele frequencies are consistently distributed in more (or less) than 0.5 in a specific genomic region. To detect such an allele imbalance event, the number of SNPs whose C57BL/6 allele frequencies are more than 0.5 was counted in each chromosome, and whether this number is significantly high or low was examined by calculating values of the cumulative density function of the binomial distribution using R. Allele imbalances are considered significant if the values are <$1 \times 10^{-8}$, which corresponds to allele imbalance involving about 50% of SNPs in a chromosome.

Analysis in the high-density GigaMUGA array was performed similarly. Briefly, among 143,259 markers in GigaMUGA array, 27,225 SNPs in autosomes are defined as informative SNPs based on consensus genotypes of two strains, and strain-specific allele frequencies were calculated in each SNP. Whether C57BL/6 allele frequencies have significant biased distribution was examined in

each chromosome by calculating values of the cumulative density function of the binomial distribution. Allele imbalances are considered significant if the values are <$1 \times 10^{-89}$, which corresponds to allele imbalance involving about 50% of SNPs in a chromosome.

**Analysis of copy number alterations**. LRR values in each sample were normalized by normalized by means and standard deviations of LRRs in autosomes to compare among different samples. Normalized LRRs in each chromosome were statistically compared with those in the other samples by paired *t*-test, and the median *p* value of each chromosome was obtained. A *p* value of <0.001 was considered as statistically significant.

**Evaluation of indel frequencies by TIDE analysis**. About 800 bp fragments around genome editing sites were amplified by PCR using following primers: (*p53*) 5′-GCTTTCCCACCCTCGGCATAA-3′ and 5′-TGTGTTCTCAGCCTACCAGC-3′, (*Pten*) 5′-GAGTCGCCTGTCACCATTGC-3′ and 5′-GTTCCGTCTAGCCGAAC ACT-3′, (*Cdkn2a*) 5′-GATTCGAACTGCGAGGACCC-3′ and 5′-CCTGCGAAGG CCAAGGTTTA-3′. PCR products purified by gel extraction were sequenced by Sanger sequencing (OHSU Vollum DNA Sequencing Core, or Genewiz, South Plainfield, NJ). Sequence chromatograms were analyzed by TIDE (https://tide. deskgen.com/)[50] to examine indel frequencies. To evaluate indel frequencies in CRISPR/Cas9-transfected cells, cells that were left on dishes after trypsinization and were not used for transplantation were harvested for TIDE analysis at 3 days after transfection.

**Quantification of contamination by real-time PCR**. DNAs extracted from tumors were analyzed by quantitative real-time PCR using FastStart Essential DNA Green Master (Roche) and LightCycler 96 System (Roche, Basel, Switzerland). Exon 5 of the *Fah* gene and *Omp* gene were amplified using following primers: (*Fah*) 5′-GGACTT CTACTCTTCTCGGCA-3′ and 5′-CAATTTGGCAACAGCGCATTC-3′, (*Omp*) 5′-GCCCACTTGATTCCCTGA-3′ and 5′-GCATCTGCTGGGTCAGGTCC-3′. As *Fah*$^{-/-}$ mice lack the target region of this primer set, only Fah-positive cells derived from donor hepatocytes were detected by this PCR assay. *Omp* gene was used as an internal control[51]. The ratio of Fah-positive cells was calculated by normalizing copy number of *Fah* gene by that of *Omp* gene.

**Statistics and reproducibility**. Statistical analysis was performed by Student's *t* test, paired *t*-test, or chi square test using R software (version 3.4.3, R Foundation for Statistical Computing), Microsoft Office Excel (Microsoft), or Prism 7 (GraphPad Software, Inc.). All representative images reflect a minimum of three biological replicates unless otherwise indicated.

**Reporting summary**. Further information on research design is available in the Nature Research Reporting Summary linked to this article.

## Data availability
Source data are provided with this paper. All the other data supporting the findings of this study are available within the article, Supplementary Information files, or from the corresponding author upon reasonable request.

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

## Acknowledgements

The AAV-Ttr-Cre construct was kindly provided by Dr. Holger Willenbring of UCSF. This work was supported by Japan Society for the Promotion of Science (JSPS) Overseas Research Fellowships, KANAE Foundation for the Promotion of Medical Science the 46th Kanae Foreign Study Grants, and National Institutes of Health grant CA190144.

## Author contributions

T.M. and M.G. conceived the project, designed the experiments, and wrote the paper. T.M. performed most of the experiments and analyzed the data. L.W. and A.P. assisted with experiments and provided technical help. M.P. and P.S. supported bioinformatic analysis of SNP array data.

## Competing interests

Oregon Health and Science University and M.G. have a significant financial interest in Yecuris, Inc. a company that may have a commercial interest in the results of this research and technology. This potential conflict of interest has been reviewed and managed by Oregon Health and Science University. The remaining authors declare no competing interests.
