## [Peer Review File · Nature Communications]

REVIEWERS' COMMENTS

Reviewer #1 (Remarks to the Author):

In this manuscript, Matsumoto et al. use a set of fluorescent reporters to investigate hepatocyte ploidy during liver regeneration and tumorigenesis. They report that polyploid hepatocytes frequently undergo ploidy reduction when actively dividing, but this capacity is lost during serial transfer. They use this observation to investigate the role of ploidy reduction in tumor development in polyploid cells. They claim that, as serially-transferred polyploid cells are more resistant to transformation in a competition assay than naive polyploid cells, ploidy reduction promotes polyploid tumorigenesis.

This manuscript includes what I think can fairly be described as an enormous amount of work, including several different mouse chromosome labeling and oncogenesis models. The central observations are important for understanding hepatocyte biology and are generally well-supported by the experiments that were conducted. I would support the publication of this manuscript in *Nature Communications* without any additional experiments, and with only a few minor changes to the text/figures.

1. In a few instances, the authors seem as though they're trying to draw a contrast between their results and the results of Zhang et al. (*Dev Cell*), who reported that polyploidy functions as a tumor suppressor in the liver. In fact, I think that these two papers are compatible: it's clear that the authors recover fewer dual-labeled tumors than expected, which likely results from both ploidy reduction and from the fact that diploid hepatocytes are more susceptible to transformation than polyploids. Accordingly, I would suggest revising the first paragraph of the discussion to portray this work as consistent with Zhang et al., rather than overthrowing it. So far as I can tell, nothing that the authors present in this current manuscript is inconsistent with the claim "hepatocyte polyploidy results in a moderate decrease in tumorigenic potential", which was my major takeaway from the Zhang study.

2. I have a problem with the sentence "Although it has been controversial whether polyploids in normal tissues generate significant aneuploidy during tissue regeneration (25-27), we here unambiguously demonstrate frequent whole-chromosome imbalances of paternal and maternal chromosomes in healthy polyploid-derived RNs, especially after ploidy reduction. This glosses over the history of this controversy and misrepresents the prior literature. In short, many researchers, including the Grompe lab, used FISH and metaphase karyotyping to claim that up to 50% of normal hepatocytes were aneuploid. Knouse et al. (ref 26) used single-cell sequencing to show that the level of aneuploidy in hepatocytes was actually <5%. Knouse et al. then followed up that study by showing that hepatocytes undergo aberrant mitoses when cultured in vitro, but such mitoses are largely absent when tissue architecture is maintained in vivo (Knouse et al., *Cell* 2018). The earlier results from Grompe and others likely represent a combination of tissue culture and FISH-related artefacts. Knouse never claimed that aneuploidy was non-existent, and Knouse did not specifically deal with regenerating liver. If the authors have a problem with the results of Knouse et al., I would suggest that they perform single-cell sequencing on freshly-isolated hepatocytes from healthy individuals and report their findings. The results included in this manuscript are in no way a refutation of the papers cited.

3. Pg 8 - "The decreased frequency of bicolored tumors in this polyploid-derived tumorigenesis model suggests that some polyploids underwent ploidy reduction during carcinogenesis." - isn't it also consistent with a model in which polyploid cells are resistant to transformation, which you go on to examine in the subsequent section?

4. Figure 1d: tdTomato is misspelled.

5. Figure 3e legend: "consistent with" is misspelled.

6. Figure S5: can the authors quantify the single-positive and double-positive cell populations?

All in all - I think that this is an important manuscript and the experiments were very thorough. I commend the authors for this piece of work, and after the modifications listed above are made, I'd fully support its publication.

Reviewer #2 (Remarks to the Author):

This is a revised manuscript that we previously reviewed for Nature. We previously had suggested an in vivo tumorigenesis experiment (see below), however, we were generally favorable about the overall strength of the manuscript. Although this experiment has not been done, our main criticisms have been addressed by more qualified discussion. We therefore support the publication of this paper in Nature Communications in its current form.

Summary of the manuscript. Matsumoto et al., address an interesting general question, how polyploidy contributes to cancer. Polyploidy occurs during the development of ~35% of human tumors and has been shown to be capable of promoting tumorigenesis in experimental models. Nevertheless, theory and microbial evolution experiments show that polyploidy has complex effects on the rate of evolutionary adaptation. Although more gene copies should increase the rate of acquiring dominant oncogenic mutations, polyploidy will buffer the loss of recessive tumor suppressor genes. Additionally, polyploidy typically is accompanied by centrosome amplification, chromosomal instability and aneuploidy, all of which may promote tumorigenesis, albeit with their own positive and negative effects. Finally, some tissues are developmentally programmed to become polyploid, such as the liver, the topic of the current manuscript. These tissues may have specific adaptations to polyploidy that shift the balance between pro- and anti- tumor effects of polyploidy. For example, polyploid liver cells, although containing extra centrosomes, appear to be less aneuploid than would otherwise be expected.

The current manuscript describes the role of polyploidy during the formation of hepatocellular carcinoma using mouse models and genomic analysis. First, and in contrast to their prior results, the authors show that in the setting of liver regeneration, polyploid cells can reduce their chromosome content, becoming aneuploid in the process. This analysis is based on a clever use of the Confetti system. There has been significant disagreement over whether polyploid hepatocytes become aneuploid during hepatic regeneration, after chronic injury, or during tumorigenesis. Here the authors use allelic ratio analysis of informative SNPs in heterozygous mice to detect aneuploidy. The analysis supports a degree of aneuploidy, consistent with the overall model, although lower than the authors previously had suggested. The allelic analysis is an advance over prior experiments measuring copy number alterations from exome sequencing, but is not as definitive as single cell sequencing would have been.

Next, they show that serial transplantation yields genetically stable polyploid cells that have lost their extra centrosomes, analogous to previous studies in cell lines. Using the RGBow system, they then show that polyploid cells robustly undergo oncogenic transformation, both after the forced expression of oncogenes and, most importantly, for the development of spontaneous tumors after hepatic injury of various types. The final series of experiments supports the conclusion that ploidy reduction (i.e chromosome loss from polyploid proliferation in stressed conditions) is required for the transformation of polyploid hepatocytes. In a compelling experiment, diploid, tetraploid and octoploid cells were used in a competitive spontaneous oncogenesis assay. This experiment showed that polyploid cells robustly underwent transformation, although at slightly reduced frequency. The fact that most tumors derived from the polyploids were monocolor, suggests that the oncogenic polyploids had undergone ploidy reduction. The final series of experiments more directly test the hypothesis that ploidy reduction is required for polyploid hepatocytes to undergo transformation. Here they compare the polyploid population that is genetically stable (derived from serial transplantation) to the ones that are not, using the competitive oncogenesis assay. In this experiment, oncogenic mutations were introduced by CRISPR. Our main criticism of

the paper is about the design of this. In principle, polyploid should prevent tumorigenesis driven by tumor suppressor loss if ploidy cannot be reduced. The authors test this but use CRISPR editing to induce tumor loss. Because CRISPR (with some variability) will eliminate most alleles simultaneously, circumventing the buffering effect of polyploidy, the experiment does not directly establish that ploidy reduction overcomes the buffering effect of polyploidy on the loss of tumor suppressors. There are some caveats that the polyploid cells that have lost their extra centrosomes could have undergone some other genetic or physiological alterations. In response to these points the authors now have softened their claims and discussed the issues fairly.

In summary, this paper addresses an important and debated issue in the literature of whether polyploid cells of the liver can generate tumors or whether in this tissue the buffering effects of polyploidy dominate. There are technical advances in this paper (e.g. the Confetti system, the direct competition experiments and the allelic copy number analysis) that support the general conclusions that (1) polyploid hepatocytes can undergo significant chromosome loss during transformation and (2) this ploidy reduction facilitates transformation (as compared to the non-reducing serially transplanted polyploid cells).

Reviewer #4 (Remarks to the Author):

The manuscript from Matsumoto et al. entitled "Proliferative polyploid cells give rise to tumors via ploidy reduction" is a timely report aiming to address an important question regarding whether polyploidisation and specifically the transformation from a polyploid to a 'ploidy reduced' state is associated with an increased risk of malignant transformation. This is an intriguing study and one which I amongst others have discussed actively following the original publication in Cell Stem Cell recently by this group using related methodology.

To study the role of ploidy reduction in liver carcinogenesis they use the loss of dual reporter expression in vivo in mice. This takes advantage of transplantation assays examining regenerative nodules and also 'transformed hepatocytes' in transplantation cancer assays. It also includes toxin mediated injury, rescue of FAH deficiency in Hgd heterozygotes as well as appropriate aging controls.

They show that imbalance with segregation at whole chromosome level may provide a selective advantage in hepatocytes, implying ploidy reduction at the chromosome level. They then show a stabilisation of ploidy in repeated transplantation experiments which relate associatively to presence of mononucleations and monocentrosomes. In HdTV (hydrodynamic tail vein) models they induce reporters together with addition of oncogenes (Akt, Myc, YAP) and show a reduced formation of tumors by polyploid hepatocytes compared to what would be predicted given the baseline transformed population. Crucially here monochrome tumors become more prevalent than what would be expected given the monochrome populations at baseline. The ploidy status of the tumors themselves was not reported however. Additionally in a "competitive oncogenesis assay" they show that polyploid cells are less likely to form tumors than their diploid counterparts but that when transplanted polyploid cancer precursors (by loss of tumour suppressors) will typically lose one of their heterozygous reporters implying ploidy reduction. Finally they show that serially transplanted "stable" polyploid hepatocytes are less tumorigenic than naive polyploid hepatocytes using 4 mice as recipients.

In general the report is written relatively clearly and concisely but may not be that approachable for the non-technical reader. It would benefit from focus on the clarity of message and might be improved by restructuring the order in which the data is presented.

Overall, the authors are to be congratulated tackling this novel question.

Major comments

In the assay where manipulated polyploid cells are transplanted in a tumor assay (Fig 4b-f) the authors show that most of the hepatocyte derived tumors lose the expression of a reporter. This is a central and particularly elegant experiment. Using sorted cells helps to remove questions in my mind related to earlier experiments about potential fusion events, for example. None the less it should be shown that the polyploid nature of these cells is retained up to the point of implantation

into the recipients and that ploidy reduction is not associated with the in vitro culture and manipulation directly. It should be shown that the tumors with loss of dual reporting are actually reduced in ploidy compared to their tetraploid and octoploid originators.

These data seem at odds with the unfractionated experiments earlier (Fig 3d) where the majority of bicolored tumors persisted as dual colored. The fact that in the competitive assays polyploid tumors are consistently less tumorigenic than their diploid counterparts (Fig4a) would argue against a selective advantage of clones that are able to undergo ploidy reduction as a driving step towards cancer in that assay. The authors appropriately discuss these interesting observations when comparing oncogenes versus tumor suppressors.

The key finding that ploidy reduction can occur and may be a source of carcinogenesis during polyploid derived cancer development is implied but crucially is not directly shown in the data presented in Figure 3 (and related extended data). Pre-labelling with Cre will induce monochrome diploid hepatocytes and mono or multicolored polyploid hepatocytes. These colored hepatocytes may also then be subsequently transfected for potential future cancer formation. The differences in multicolored tumors vs. hepatocytes at baseline (shown in Fig 3d), therefore, does not prove that polyploids have undergone ploidy reduction. This may also be selective advantage of biploid progenitors to grow out (or other mechanisms, including evasion of clearance) over their polyploid counterparts. The data presented would be consistent with their hypothesis but does not rule out other explanations. It is important to note that, whilst I agree with the schematics in Sup Fig 6e,f; as the ploidy status was not measured in the tumors these schematics are not directly comparable to the data presented in related Figure 3 and are therefore somewhat misleading. Sorting out polyploid multi-colored clones and then examining the ploidy status of the resulting tumors would clinch this.

Similar arguments about the cell of origin and timing of original labelling in the chronic injury induced cancer models (3f-h). These models are less elegant in my opinion than the co-registration of lineage and oncogene in the earlier experiments (Fig 3a-e).

Figure 1 could be restructured to more clearly and succinctly demonstrate the point about chromosome mis-segregation. Currently the balance between data in Supp Fig.1 and Fig 1 is not right in my opinion. I do not find this easy to follow given the descriptions provided. Furthermore the title of the first section should highlight that the regenerative nodules are specific to this transplantation model (rather than what most consider to be regenerative nodules in disease) and that ploidy reduction does not somehow promote regenerative nodule formation. The data in Figure 2b in the 2nd recipients argues against selective advantage between monochrome clones versus the polychrome multiploids. The key data showing selective chromosome segregation in the elegant rescue experiment in Ext Fig 3 would be worth of a place in a main figure in my opinion.

It is not clear how the data in Fig 5e is generated. Was each tumor sequenced individually and if so was there a correlation between those tumors that did lose p53 and loss of reporter compared to those that did not lose p53.

Data should be provided to validate efficient PTEN and p53 editing in both diploid and polyploid cells.

I am concerned about the expression of Cre in the cancer models. It isn't clear from the wording whether this model also uses a transposon system and is Cre recombinase therefore also integrated. I assume it is not but if it is, then there would be long term Cre expression which might affect stability of the reporter.

Minor comments

Line 61 "could be the origin of cancers." – I would suggest changing to could be the origin of some cancers.

For the SNP analysis in Figure 1 in the uninjured state it would be interesting to segregate chromosome 6 (based on its possession of the Rosa locus) with the parental strain for the

reporter, if this is possible given the experimental set up. It would be anticipated that (providing there was balance at the time of original cell harvest) that you would see preferential AI segregation of any single or multicolored RNs. This is not essential but would again act to support the authors' claims.

For labelling of 1st and 3rd transplant in Figure 2 it would be helpful if the same terminology could be applied to Fig 2e,f. Similarly the methodology for quantifying mononucleation versus multinucleation should be stated. Additionally can examples of immunostaining for centrosomes be shown in liver sections to ensure that the quantification performed in vitro is not an artefact of plating overnight between transplanted and serially transplanted hepatocytes?

It would be interesting to examine or speculate on why there is specific allelic imbalance in chromosome 9. Is this a feature of the FAH transplant model or a more general selective advantage between the mouse backgrounds?

I am not clear whether the transposon system delivers plasmids predominantly into polyploids or whether cells which have had then delivered into them become polyploid in response to transfection. This is semantic but should be clarified in the text (line 157).

Citation 6 comes after 7,8 and 9, so numbers should be switched

In the discussion (line 294/295) is a callout to Ext. Fig 11 c-e, is this Ext Fig 10 c-e?

In Figure 4d there's an "o" missing

Ext Figure 1, should Ext Fig 1. d and e be labelled the other way around?

The comment regarding tumor evolution and ploidy reduction is rather speculative (line 213). This could be discussed further. What was the number of tumors that had that dual phenotype and is it possible that these were from two separate induction originators

The mechanism for stabilisation of ploidy reduction suppression after prolonged proliferation is clearly highly interesting and worthy of ongoing future research.

Reviewer #4 (Remarks to the Author):

Addressing reviewer's comments raised by reviewer #3

The concerns raised by this reviewer (and others, particularly reviewer #1) are broadly in line with my own independent review which was made without prior access to the other reviewers comments. I have therefore specifically revisited reviewer #3 concerns, but additionally highlight my opinion on the rebuttal of reviewer #1 comments additionally (italicised) should this is of interest to the editorial team.

Reviewer #3

1. Concern regarding the different models used and the lack of demonstration of enhanced tumorigenesis specifically from polyploid hepatocytes.

As outlined in my own independent review of the manuscript (prior to receipt of previous reviewer's comments) I share this concern in the most recent version of the manuscript. Assessing the rebuttal I entirely agree regarding the authors' hypothesis regarding susceptibility to LOH (incidentally a point which is not clearly made in the current manuscript. The explanation of the experimental challenges for in vivo manipulation of transplanted hepatocytes are reasonable but the fundamental problem of lack of demonstration of enhanced tumorigenesis by this population remains. Therefore, I do not believe this rebuttal adequately addresses the concerns raised by the reviewer on this matter.

2. Regarding the changes in frequencies based on monochrome versus multicolored tumors in revised Fig 3D and 3H.

Again as outlined in my own independent review of the most recent manuscript I have ongoing concerns regarding the assumptions drawn from this data. I can comment on the data presented in the current version of the manuscript which has been modified in an attempt to address the reviewer's concern. I understand the rebuttal points made but my concerns still exist and I would highlight that the analysis shown in theory in Extended Fig 6e/f are different to the analyses used in Fig 3D and 3H. Therefore, on balance I do not feel that the rebuttal adequately addresses the concern raised on this point by the reviewer.

3. Concern regarding novelty related to ploidy reduction

I agree with the authors' rebuttal to this point which satisfactorily addresses the concerns of the reviewers. I would highlight that for the non-specialist reader the current version of the manuscript may be challenging with respect to these issues.

4. Evaluation of relevance of ploidy reduction in greater detail.

The reviewer raises a relevant point, however, I would agree with the authors that while future investigation of this characterisation is of scientific merit it is not required at this time for the fundamental description of the importance of ploidy reduction in carcinogenesis.

5. Characterisation of ploidy and nuclear organization of hepatocytes

The authors' rebuttal regarding characterisation of this in their previous publication is fair. However my reading of this concern is regarding the characterisation of the cells immediately pre transplantation and post transplantation from the tumors themselves. This analysis would be beneficial in my view.

I accept the authors' observation for bias between the YFP and RFP monochrome cells, although appreciate why this remains untested.

6. Concerns regarding clumping of cells, especially over time, in vitro prior to sorting

These technical concerns will remain with, as the authors' concede, the inevitable impurity of sorted populations. I appreciate the concerns of the reviewer but overall would agree with the authors that they have suitably (both here and previously) addressed this concern to the best of what is reasonably achievable.

Minor points

1. Quantification of regenerative nodules

Rebuttal accepted

2. Description of experimental technical details

Rebuttal accepted

3. Concern regarding rescue of FAH deficiency experiment (Ext Fig 3)

Personally, I found this an elegant experiment which, as the authors' report, was designed to show that chromosomal aberrations in situ. I accept the authors' rebuttal of this point.

4. Mechanism of reduced ploidy reduction after serial transplantation

The authors highlight the ongoing proliferative potential of serially transplanted hepatocytes in this model based on published data. This point is fair. I would go further than this query made by the reviewer and question whether the effects upon serially transplantation affect tumorigenic potential in ways other than the ability to undergo ploidy reduction as highlighted also by reviewer #1. In answer to the reviewers questions the rebuttal is sufficient, however in order to address concerns regarding experimental validity I remain with concern in this area despite the reassurances provided in the rebuttal.

5. Comparison to baseline transplantation in the serial transplantation studies.

Rebuttal accepted.

6. Presentation of serially transplanted hepatocyte images

Rebuttal accepted

Reviewer #1

1 A) Novelty.

I would agree with the authors' rebuttal that the field is currently undecided regarding the role or not of polyploidy with relevance to hepatocellular carcinoma. I would see this as distinct to polyploidy as the liver is an organ where polyploidy is the norm physiologically. I agree with the authors' rebuttal and do not see lack of novelty as a preclusion to publication.

1. *B) Alternative explanation for the observations made.*

I agree with the reviewers point here which is consistent with my concerns raised in my independent review. While the experimental data is supportive it does not elegantly prove their hypothesis. This issue remains unresolved in the most recent submission. I agree with the previous reviewer that a rescue experiment restoring reductive mitosis in the serially transferred cells would be an excellent way to address this. Whilst I accept the arguments made in rebuttal to this they do not adequately address the concerns raised by the previous reviewer.

2. *Loading in favor of polyploids in the use of tumor suppressors.*

Whilst I see the reviewers point I think that the combination of models using hyperexpression of oncogenes Myc/YAP/Akt in addition to deletion of tumor suppressors p53/PTEN and p16/ARF is a reasonable approach. Obviously the group have attempted additional yet unsuccessful strategies. Therefore on balance I am reassured by the authors' rebuttal and discussion on this issue specifically.

3. *Crispr in efficiency related to point 1B above; in serially transplanted hepatocytes.*

This is a very good suggestion made by the reviewer which had not occurred to me, and again would be a confounder in the interpretation. This comes back to the validation of the Crispr efficiency. Again this issue would be reconciled by a successful rescue experiment. The authors' rebuttal regarding equal treatment of cells does not equate to equal response with the Crispr method. The equivalent variation between octaploid and tetraploid (measured by TIDE) additionally does not reassure as the key comparison here is to diploids not between polyploids. The rebuttal goes some way to addressing this point but does not fully reassure and the concerns raised relate to point 1B above remain.

Minor points

1. *Rebuttal accepted*
2. *Rebuttal accepted*
3. *Rebuttal accepted*
4. *Rebuttal accepted*
5. *This is an interesting question but I would side with the authors that the SNP array (to my understanding at least) would not address this point. I do not see addressing this point as critical to the current manuscript.*

Responses to the Reviewers' Comments:

We thank the reviewers for their thoughtful comments, which have helped us to improve our paper. We responded to each of the reviewer's comments through the following changes to the manuscript and will address each reviewer comment one by one. Changes are shown in red in the revised paper. We also revised Figure 1 and Extended data figure S1. In addition, we made slight modifications in the labels of supplementary figures to follow the guidelines of the journal, and corrected trivial errors in Supplementary Figs. 10a and 10b.

Reviewer #1:

In this manuscript, Matsumoto et al. use a set of fluorescent reporters to investigate hepatocyte ploidy during liver regeneration and tumorigenesis. They report that polyploid hepatocytes frequently undergo ploidy reduction when actively dividing, but this capacity is lost during serial transfer. They use this observation to investigate the role of ploidy reduction in tumor development in polyploid cells. They claim that, as serially-transferred polyploid cells are more resistant to transformation in a competition assay than naive polyploid cells, ploidy reduction promotes polyploid tumorigenesis.

This manuscript includes what I think can fairly be described as an enormous amount of work, including several different mouse chromosome labeling and oncogenesis models. The central observations are important for understanding hepatocyte biology and are generally well-supported by the experiments that were conducted. I would support the publication of this manuscript in Nature Communications without any additional experiments, and with only a few minor changes to the text/figures.

1. In a few instances, the authors seem as though they're trying to draw a contrast between their results and the results of Zhang et al. (Dev Cell), who reported that polyploidy functions as a tumor suppressor in the liver. In fact, I think that these two papers are compatible: it's clear that the authors recover fewer dual-labeled tumors than expected, which likely results from both ploidy reduction and from the fact that diploid hepatocytes are more susceptible to transformation than polyploids. Accordingly, I would suggest revising the first paragraph of the discussion to portray this work as consistent with Zhang et al., rather than overthrowing it. So far as I can tell, nothing that the authors present in this current manuscript is inconsistent with the claim "hepatocyte polyploidy results in a moderate decrease in tumorigenic potential", which was my major takeaway from the Zhang study.

Reply:

We wish to express our strong appreciation for your insightful comments on our paper. We feel the comments have helped us significantly improve the paper.

According to the reviewer's comment, we revised the first paragraph of the Discussion by clearly describing that our results were consistent with the previous study (Zhang S et al. Dev Cell, 2018), and that we advanced insights into polyploid-derived tumorigenesis especially by showing ploidy reduction (page 14, lines 265-269).

2. I have a problem with the sentence "Although it has been controversial whether polyploids in normal tissues generate significant aneuploidy during tissue regeneration (25-27), we here

unambiguously demonstrate frequent whole-chromosome imbalances of paternal and maternal chromosomes in healthy polyploid-derived RNs, especially after ploidy reduction. This glosses over the history of this controversy and misrepresents the prior literature. In short, many researchers, including the Grompe lab, used FISH and metaphase karyotyping to claim that up to 50% of normal hepatocytes were aneuploid. Knouse et al. (ref 26) used single-cell sequencing to show that the level of aneuploidy in hepatocytes was actually <5%. Knouse et al. then followed up that study by showing that hepatocytes undergo aberrant mitoses when cultured in vitro, but such mitoses are largely absent when tissue architecture is maintained in vivo (Knouse et al., Cell 2018). The earlier results from Grompe and others likely represent a combination of tissue culture and FISH-related artefacts. Knouse never claimed that aneuploidy was non-existent, and Knouse did not specifically deal with regenerating liver. If the authors have a problem with the results of Knouse et al., I would suggest that they perform single-cell sequencing on freshly-isolated hepatocytes from healthy individuals and report their findings. The results included in this manuscript are in no way a refutation of the papers cited.

Reply:

We totally agree that aneuploidies that we analyzed were in regenerative nodules, and Knouse et al. examined hepatocytes from healthy livers. According to the reviewer's comment, we revised the sentence that the reviewer pointed out (page 14, lines 283-289).

3. Pg 8 - "The decreased frequency of bicolored tumors in this polyploid-derived tumorigenesis model suggests that some polyploids underwent ploidy reduction during carcinogenesis." - isn't it also consistent with a model in which polyploid cells are resistant to transformation, which you go on to examine in the subsequent section?

Reply:

We agree that the decreased frequency of bicolored tumors compared to the baseline can also be explained if polyploid cells are resistant to transformation. According to the reviewer's comment, we modified the sentence that the reviewer pointed out avoiding a misleading description (page 8, lines 169-172).

4. Figure 1d: tdTomato is misspelled.

Reply:

We apologize for this oversight and have made corrections.

5. Figure 3e legend: "consistent with" is misspelled.

Reply:

This error in Figure 3e legend has been corrected in accordance with the reviewer's comment.

6. Figure S5: can the authors quantify the single-positive and double-positive cell populations?

Reply:

According to the reviewer's comment, we quantified the frequencies of ploidy reduction in the context of CCl4-liver injury, and confirmed that naïve hepatocytes frequently gave rise to clones with ploidy reduction ($119/173 = 68.8\%$), while serially-transplanted hepatocytes did not undergo ploidy reduction ($0/127 = 0\%$). We added these quantitative data to Supplementary Fig. 5 of the revised manuscript.

All in all - I think that this is an important manuscript and the experiments were very thorough. I commend the authors for this piece of work, and after the modifications listed above are made, I'd fully support its publication.

Reviewer #2:

This is a revised manuscript that we previously reviewed for Nature. We previously had suggested an *in vivo* tumorigenesis experiment (see below), however, we were generally favorable about the overall strength of the manuscript. Although this experiment has not been done, our main criticisms have been addressed by more qualified discussion. We therefore support the publication of this paper in Nature Communications in its current form.

Summary of the manuscript. Matsumoto et al., address an interesting general question, how polyploidy contributes to cancer. Polyploidy occurs during the development of ~35% of human tumors and has been shown to be capable of promoting tumorigenesis in experimental models. Nevertheless, theory and microbial evolution experiments show that polyploidy has complex effects on the rate of evolutionary adaptation. Although more gene copies should increase the rate of acquiring dominant oncogenic mutations, polyploidy will buffer the loss of recessive tumor suppressor genes. Additionally, polyploidy typically is accompanied by centrosome amplification, chromosomal instability and aneuploidy, all of which may promote tumorigenesis, albeit with their own positive and negative effects. Finally, some tissues are developmentally programmed to become polyploid, such as the liver, the topic of the current manuscript. These tissues may have specific adaptations to polyploidy that shift the balance between pro- and anti- tumor effects of polyploidy. For example, polyploid liver cells, although containing extra centrosomes, appear to be less aneuploid than would otherwise be expected.

The current manuscript describes the role of polyploidy during the formation of hepatocellular carcinoma using mouse models and genomic analysis. First, and in contrast to their prior results, the authors show that in the setting of liver regeneration, polyploid cells can reduce their chromosome content, becoming aneuploid in the process. This analysis is based on a clever use of the Confetti system. There has been significant disagreement over whether polyploid hepatocytes become aneuploid during hepatic regeneration, after chronic injury, or during tumorigenesis. Here the authors use allelic ratio analysis of informative SNPs in heterozygous mice to detect aneuploidy. The analysis supports a degree of aneuploidy, consistent with the overall model, although lower than the authors previously had suggested. The allelic analysis is an advance over prior experiments measuring copy number alterations from exome sequencing, but is not as definitive as single cell sequencing would have been.

Next, they show that serial transplantation yields genetically stable polyploid cells that have lost their extra centrosomes, analogous to previous studies in cell lines. Using the RGBow system, they then show that polyploid cells robustly undergo oncogenic transformation, both

after the forced expression of oncogenes and, most importantly, for the development of spontaneous tumors after hepatic injury of various types. The final series of experiments supports the conclusion that ploidy reduction (i.e chromosome loss from polyploid proliferation in stressed conditions) is required for the transformation of polyploid hepatocytes. In a compelling experiment, diploid, tetraploid and octoploid cells were used in a competitive spontaneous oncogenesis assay. This experiment showed that polyploid cells robustly underwent transformation, although at slightly reduced frequency. The fact that most tumors derived from the polyploids were monocolor, suggests that the oncogenic polyploids had undergone ploidy reduction. The final series of experiments more directly test the hypothesis that ploidy reduction is required for polyploid hepatocytes to undergo transformation. Here they compare the polyploid population that is genetically stable (derived from serial transplantation) to the ones that are not, using the competitive oncogenesis assay. In this experiment, oncogenic mutations were introduced by CRISPR. Our main criticism of the paper is about the design of this. In principle, polyploid should prevent tumorigenesis driven by tumor suppressor loss if ploidy cannot be reduced. The authors test this but use CRISPR editing to induce tumor loss. Because CRISPR (with some variability) will eliminate most alleles simultaneously, circumventing the buffering effect of polyploidy, the experiment does not directly establish that ploidy reduction overcomes the buffering effect of polyploidy on the loss of tumor suppressors. There are some caveats that the polyploid cells that have lost their extra centrosomes could have undergone some other genetic or physiological alterations. In response to these points the authors now have softened their claims and discussed the issues fairly.

In summary, this paper addresses an important and debated issue in the literature of whether polyploid cells of the liver can generate tumors or whether in this tissue the buffering effects of polyploidy dominate. There are technical advances in this paper (e.g. the Confetti system, the direct competition experiments and the allelic copy number analysis) that support the general conclusions that (1) polyploid hepatocytes can undergo significant chromosome loss during transformation and (2) this ploidy reduction facilitates transformation (as compared to the non-reducing serially transplanted polyploid cells).

Reply:

We wish to express our strong appreciation for your insightful comments on our paper. In particular, we feel that your previous review for Nature have helped us significantly improve the paper. We sincerely appreciate the time and energy you expended towards improving our manuscript.

Reviewer #4:

The manuscript from Matsumoto et al. entitled “Proliferative polyploid cells give rise to tumors via ploidy reduction” is a timely report aiming to address an important question regarding whether polyploidization and specifically the transformation from a polyploid to a ‘ploidy reduced’ state is associated with an increased risk of malignant transformation. This is an intriguing study and one which I amongst others have discussed actively following the original publication in Cell Stem Cell recently by this group using related methodology.

To study the role of ploidy reduction in liver carcinogenesis they use the loss of dual reporter expression in vivo in mice. This takes advantage of transplantation assays examining regenerative nodules and also ‘transformed hepatocytes’ in transplantation cancer assays. It also includes toxin mediated injury, rescue of FAH deficiency in Hgd heterozygotes as well as appropriate aging controls.

They show that imbalance with segregation at whole chromosome level may provide a selective advantage in hepatocytes, implying ploidy reduction at the chromosome level. They then show a stabilization of ploidy in repeated transplantation experiments which relate associatively to presence of mononucleations and monocentrosomes. In HdTV (hydrodynamic tail vein) models they induce reporters together with addition of oncogenes (Akt, Myc, YAP) and show a reduced formation of tumors by polyploid hepatocytes compared to what would be predicted given the baseline transformed population. Crucially here monochrome tumors become more prevalent than what would be expected given the monochrome populations at baseline. The ploidy status of the tumors themselves was not reported however. Additionally in a “competitive oncogenesis assay” they show that polyploid cells are less likely to form tumors than their diploid counterparts but that when transplanted polyploid cancer precursors (by loss of tumour suppressors) will typically lose one of their heterozygous reporters implying ploidy reduction. Finally they show that serially transplanted “stable” polyploid hepatocytes are less tumorigenic than naïve polyploid hepatocytes using 4 mice as recipients.

In general the report is written relatively clearly and concisely but may not be that approachable for the non-technical reader. It would benefit from focus on the clarity of message and might be improved by restructuring the order in which the data is presented.

Overall, the authors are to be congratulated tackling this novel question.

Reply:

We wish to express our strong appreciation for your insightful comments on our paper. We feel the comments have helped us significantly improve the paper.

According to the reviewer’s comment, we restructured Figure 1 to make it clearer as described later. We also responded to the reviewer’s comment on the ploidy status of tumors in the following

major points.

Major comments

In the assay where manipulated polyploid cells are transplanted in a tumor assay (Fig 4b-f) the authors show that most of the hepatocyte derived tumors lose the expression of a reporter. This is a central and particularly elegant experiment. Using sorted cells helps to remove questions in my mind related to earlier experiments about potential fusion event, for example. None the less it should be shown that the polyploid nature of these cells is retained up to the point of implantation into the recipients and that ploidy reduction is not associated with the in vitro culture and manipulation directly. It should be shown that the tumors with loss of dual reporting are actually reduced in ploidy compared to their tetraploid and octoploid originators.

Reply:

We agree with the importance of proving this point. In our previous paper (Matsumoto T, et al. Cell Stem Cell 2020, Figs. 1D, E, 3B and Fig. 6.), we carefully and clearly proved that bicolored labeling indicates polyploidy, and that loss of a reporter corresponds to ploidy reduction. This was done by flow cytometry (Figs. 1D, E and 3B of the CSC paper) as well as image cytometry by microscopy on tissue sections (Fig. 6C of the CSC paper). Since we performed and reported all of these controls in our prior paper, we did not repeat them here.

It is important to note that diploid hepatocytes derived from polyploid hepatocytes via ploidy reduction readily re-polyploidize during subsequent proliferation, and thus the majority of monocolor hepatocytes derived from bicolored cells are polyploid in the recipient liver after repopulation (Matsumoto T, et al. Cell Stem Cell 2020). Therefore, in our previous paper, we showed ploidy reduction and re-polyploidization by sorting diploid cells from the first recipient *Fah^{-/-}* mice as well as by image cytometric analysis in wild-type mice injured with CCl₄ (Matsumoto T, et al. Cell Stem Cell 2020). As this re-polyploidization would be confusing to uninitiated readers, we focused on the clarity of message and did not include ploidy data of repopulated livers in the manuscript.

In addition, we also previously demonstrated ploidy reduction in the livers without transplantation (Matsumoto T, et al. Cell Stem Cell 2020). Furthermore, the fact that ploidy reduction was observed without ex vivo culture or oncogenic manipulations in naïve hepatocytes, while serially-transplanted cells did not undergo ploidy reduction under the same condition (Figure 2c) supports that ploidy reduction is not associated with experimental manipulations.

These data seem at odds with the unfractionated experiments earlier (Fig 3d) where the majority of bicolored tumors persisted as dual colored. The fact that in the competitive assays

polyploid tumors are consistently less tumorigenic than their diploid counterparts (Fig4a) would argue against a selective advantage of clones that are able to undergo ploidy reduction as a driving step towards cancer in that assay. The authors appropriately discuss these interesting observations when comparing oncogenes versus tumor suppressors.

Reply:

As the reviewer pointed out, oncogene-induced cancer models (Fig. 3d) exhibit less frequent ploidy reduction than LOH-induced cancer models, and we discussed the significance of ploidy reduction in different kinds of tumorigenic mechanisms based on that out (pages 16-17, lines 310-321). We also agree that diploids, which don't undergo ploidy reduction, are moderately more tumorigenic than polyploids, and that ploidy reduction is not universally essential for tumorigenesis. We explicitly described this in the revised manuscript (pages 16, lines 307-309, pages 17, lines 317-319).

The key finding that ploidy reduction can occur and may be a source of carcinogenesis during polyploid derived cancer development is implied but crucially is not directly shown in the data presented in Figure 3 (and related extended data). Pre-labelling with Cre will induce monochrome diploid hepatocytes and mono or multicolored polyploid hepatocytes. These colored hepatocytes may also then be subsequently transfected for potential future cancer formation. The differences in multicolored tumors vs. hepatocytes at baseline (shown in Fig 3d), therefore, does not prove that polyploids have undergone ploidy reduction. This may also be selective advantage of diploid progenitors to grow out (or other mechanisms, including evasion of clearance) over their polyploid counterparts. The data presented would be consistent with their hypothesis but does not rule out other explanations. It is important to note that, whilst I agree with the schematics in Sup Fig 6e,f; as the ploidy status was not measured in the tumors these schematics are not directly comparable to the data presented in related Figure 3 and are therefore somewhat misleading. Sorting out polyploid multi-colored clones and then examining the ploidy status of the resulting tumors would clinch this.

Reply:

We totally agree that Supplementary Figs. 6e and 6f are not based on actual ploidy data of tumors and could be somewhat misleading. According to the reviewer's comment, we modified Supplementary Figs. 6e and 6f, and clearly indicated that the models shown in the Supplementary figure are based on theoretical deduction, not actual measurements.

As described in the first major point, ploidy-reduced hepatocytes are likely to readily re-polyploidize. Moreover, liver tumor cells are frequently polyploidized at a variety of frequency in tumors (Bou-Nader M, et al. Gut 2020). Thus, all hepatocyte fractions (monocolored diploids, monocolored polyploids, and bicolored polyploids) can give rise to monocolored tumors containing

diploids and polyploids via polyploidization and/or ploidy reduction, and it is unfortunately difficult to deduce tumor origins using ploidy status of established tumors. The only thing we can confirm with certainty is that bicolored tumors must be derived from polyploid cells.

Similar arguments about the cell of origin and timing of original labelling in the chronic injury induced cancer models (3f-h). These models are less elegant in my opinion than the co-registration of lineage and oncogene in the earlier experiments (Fig 3a-e).

Reply:

As described in the previous point, we cannot deduce tumor origins based on final ploidy status of tumors induced by chronic injury either. We have made this clear in our revised manuscript (page43, lines 855-858).

Figure 1 could be restructured to more clearly and succinctly demonstrate the point about chromosome mis-segregation. Currently the balance between data in Supp Fig.1 and Fig 1 is not right in my opinion. I do not find this easy to follow given the descriptions provided. Furthermore the title of the first section should highlight that the regenerative nodules are specific to this transplantation model (rather than what most consider to be regenerative nodules in disease) and that ploidy reduction does not somehow promote regenerative nodule formation. The data in Figure 2b in the 2nd recipients argues against selective advantage between monochrome clones versus the polychrome mutliploids. The key data showing selective chromosome segregation in the elegant rescue experiment in Ext Fig 3 would be worth of a place in a main figure in my opinion.

Reply:

According to the reviewer's comment, we restructured Fig. 1 and Supplementary Fig. 1 in the revised manuscript. We also modified the title of the first section of the Results to indicate that the data were mainly demonstrated by repopulation nodules in the transplantation model (page 5, lines 90-91). In order to make it clear that ploidy reduction promotes allele imbalances as a central message, we kept the data in Supplementary Fig. 3 as supplementary information although we agree that the rescue of liver damage by allele imbalance was interesting.

It is not clear how the data in Fig 5e is generated. Was each tumor sequenced individually and if so was there a correlation between those tumors that did loose p53 and loss of reporter compared to those that did not loose p53.

Reply:

We infer that the reviewer is mentioning Figure 4e instead of Figure 5e. In Figure 4e, each dot is derived from one recipient mouse. The x axis indicates p53 knockout frequencies in transplanted

hepatocytes and the y axis indicates frequencies of tumors losing reporter expression(s) in each mouse (17.8 tumors were analyzed in each mouse on average). Unfortunately, we didn't analyze sequences of individual tumors related to Figure 4e. Instead, we analyzed *Pten* sequences of some tumors in Supplementary Fig. 10, and showed the correlation between ploidy reduction and variability in the kinds of *Pten* indels in tumors.

Data should be provided to validate efficient PTEN and p53 editing in both diploid and polyploid cells.

Reply:

In the competitive analysis shown in Figure 4a, we mixed sorted diploid and polyploid cells at first, and transfected CRISPR/Cas9 in a mixture to ensure that each ploidy fraction is treated with the same conditions. Thus, unfortunately we don't have data about the frequencies of gene editing in each ploidy fraction originating from this experiment. Overall, indels in *p53* and *Pten* were efficiently (> 80%) induced in this experiment.

On the other hand, we separately transfected CRISPR/Cas9 into sorted diploids, tetraploids, and octaploids several times, and comparison among cells sorted on the same day demonstrated that transfection of CRISPR/Cas9 efficiently induced *p53* and *Pten* editing in each ploidy fraction. Although there was a tendency towards a higher indel frequency in diploids, the difference was minor.

I am concerned about the expression of Cre in the cancer models. It isn't clear from the wording whether this model also uses a transposon system and is Cre recombinase therefore also integrated. I assume it is not but if it is, then there would be long term Cre expression which might affect stability of the reporter.

Reply:

The transposon system was never used to induce the expression of Cre recombinase throughout the manuscript. Even in the models by hydrodynamic injection in Figures 3a-e, a conventional non-integrating plasmid was used for Cre expression as described in Materials and methods.

In all experiments in the manuscript, the expression of Cre recombinase were induced by either injection of circular non-transposon plasmids, infection of AAV-Ttr-Cre or tamoxifen in Ubc-CreERT2 mice. Once cells proliferate by transformation and/or transplantation, plasmids or AAV are soon washed out and episomal expression of Cre recombinase is lost. In addition, even if Cre proteins persistently existed in cells for a while, the design of Rosa-Confetti makes it nearly impossible to re-recombine a YFP-allele into an RFP-allele (and vice versa). Thus, problems caused

by prolonged Cre expression can be excluded in our experiments.

Minor comments

Line 61 “could be the origin of cancers.” – I would suggest changing to could be the origin of some cancers.

Reply:

Thank you. We modified the description according to the reviewer's comment.

For the SNP analysis in Figure 1 in the uninjured state it would be interesting to segregate chromosome 6 (based on its possession of the Rosa locus) with the parental strain for the reporter, if this is possible given the experimental set up. It would be anticipated that (providing there was balance at the time of original cell harvest) that you would see preferential AI segregation of any single or multicolored RNs. This is not essential but would again act to support the authors' claims.

Reply:

We appreciate the reviewer's suggestion, and agree that additional important insights would be revealed by analyzing allele imbalances based on the pairs of parental strains. Unfortunately, however, all regenerative nodules analyzed were originated from hepatocytes harvested from livers in mice with Ubc-CreERT2/Rosa-Confetti^{+/-} C57BL/6 fathers and wild-type 129S4/SvJae mothers. We would like to address it in the future research.

For labelling of 1st and 3rd transplant in Figure 2 it would be helpful if the same terminology could be applied to Fig 2e,f. Similarly the methodology for quantifying mononucleation versus multinucleation should be stated. Additionally can examples of immunostaining for centrosomes be shown in liver sections to ensure that the quantification performed in vitro is not an artefact of plating overnight between transplanted and serially transplanted hepatocytes?

Reply:

In Figs. 2e and 2f, serially-transplanted hepatocytes were collected from secondary or tertiary recipient mice by sorting bicolored polyploid cells because they both exhibit loss of ploidy reduction and similar profiles of nuclear and centrosome numbers. According to the reviewer's comment, we indicated the origins of serially-transplanted cells in the legend of Fig. 2 of the revised manuscript (page 36, lines 715-717). In addition, in accordance with the reviewer's comment, we added a description about the methodology for nuclear number quantification in the Materials and Methods of the revised manuscript (page 22, lines 440-441). We also tried to evaluate the numbers of hepatocyte centrosomes using liver sections, but unfortunately, we could not reliably do it due to the

difficulty to discriminate true signals from noise and to avoid the potential bias caused by analyzing sectional images of cells.

It would be interesting to examine or speculate on why there is specific allelic imbalance in chromosome 9. Is this a feature of the FAH transplant model or a more general selective advantage between the mouse backgrounds?

Sorry, but we do not quite understand this comment, as we do not see enrichment of allelic imbalances of chromosome 9 in our data.

I am not clear whether the transposon system delivers plasmids predominantly into polyploids or whether cells which have had then delivered into them become polyploid in response to transfection. This is semantic but should be clarified in the text (line 157).

Reply:

Hydrodynamic injection mainly delivers plasmids into hepatocytes at pericentral regions (Suda T, et al. Gene Therapy, 2007), where polyploid hepatocytes are enriched (Tanami S, et al. Cell Tissue Res, 2017). Thus, we speculate that plasmids were predominantly delivered into polyploids by hydrodynamic injection. However, we cannot completely exclude the possibility that some hepatocytes with plasmids polyploidized in response to hydrodynamic injection before the analysis at 2 or 3 days after injection. In either case, we confirmed that hepatocytes transduced by plasmids were polyploid, indicating that hydrodynamic injection served as polyploid-biased tumorigenesis models.

According to the reviewer's comment, we modified the sentence about hydrodynamic injection and polyploidy in the revised manuscript (page 8, lines 156-159).

Citation 6 comes after 7,8 and 9, so numbers should be switched

Reply:

Reference #6 was cited in the abstract of the original manuscript. According to the journal format, we removed the citations in the abstract and modified the citation numbers accordingly.

In the discussion (line 294/295) is a callout to Ext. Fig 11 c-e, is this Ext Fig 10 c-e?

Reply:

We apologize for this oversight and have made corrections.

In Figure 4d there's an "o" missing

Reply:

This error in Figure 4d has been corrected in accordance with the reviewer's comment.

Ext Figure 1, should Ext Fig 1. d and e be labelled the other way around?

Reply:

We apologize for this oversight and have made corrections.

The comment regarding tumor evolution and ploidy reduction is rather speculative (line 213). This could be discussed further. What was the number of tumors that had that dual phenotype and is it possible that these were from two separate induction originators

Reply:

Two tumors with cholangiocellular transdifferentiation were found among all 30 tumors with intratumoral ploidy reduction ($2/30 = 6.7\%$). Notably, transdifferentiation was observed at the border between a bicolored tumor portion and a monocolored one in both tumors (Fig. 4h and Supplementary Fig. 8). As both tumors contain bicolored tumor cells, they are supposed to be derived from a single bicolored origin cell. In accordance with the reviewer's comment, we added detailed information and discussion about tumors with cholangiocellular transdifferentiation in the revised manuscript (page 11, lines 214-221).

The mechanism for stabilization of ploidy reduction suppression after prolonged proliferation is clearly highly interesting and worthy of ongoing future research.

Reply:

We appreciate the reviewer's suggestion, and would like to address it in the future research.

Appendix: This is an addendum to the original rebuttal letter.

In the following are reviewer #4's comments about our original rebuttals of the comments by reviewers #1-3's.

Addressing reviewer's comments raised by reviewer #3

The concerns raised by this reviewer (and others, particularly reviewer #1) are broadly in line with my own independent review which was made without prior access to the other reviewers comments. I have therefore specifically revisited reviewer #3 concerns, but additionally highlight my opinion on the rebuttal of reviewer #1 comments additionally (italicised) should this is of interest to the editorial team.

Reviewer #3

1. Concern regarding the different models used and the lack of demonstration of enhanced tumorigenesis specifically from polyploid hepatocytes.

As outlined in my own independent review of the manuscript (prior to receipt of previous reviewer's comments) I share this concern in the most recent version of the manuscript. Assessing the rebuttal I entirely agree regarding the authors' hypothesis regarding susceptibility to LOH (incidentally a point which is not clearly made in the current manuscript. The explanation of the experimental challenges for in vivo manipulation of transplanted hepatocytes are reasonable but the fundamental problem of lack of demonstration of enhanced tumorigenesis by this population remains. Therefore, I do not believe this rebuttal adequately addresses the concerns raised by the reviewer on this matter.

Reply:

We agree that our data do not show enhanced tumorigenesis over diploids and do not claim this in our manuscript. However, the data also show that polyploids are nearly as susceptible to oncogenesis as diploids and are not significantly protected by their increased chromosome number. In response to the original review and this review we modified the discussion to directly address the significance of ploidy reduction in different kinds of tumorigenic mechanisms (page 16 of the revised manuscript). We also agree that ploidy reduction is not universally essential for tumorigenesis, and that diploids, which don't undergo ploidy reduction, are moderately more tumorigenic than polyploids (Fig. 4a). We explicitly pointed this out in the revised manuscript (page 14).

2. Regarding the changes in frequencies based on monochrome versus multicolored tumors in revised Fig 3D and 3H.

Again as outlined in my own independent review of the most recent manuscript I have ongoing concerns regarding the assumptions drawn from this data. I can comment on the data

presented in the current version of the manuscript which has been modified in an attempt to address the reviewer's concern. I understand the rebuttal points made but my concerns still exist and I would highlight that the analysis shown in theory in Extended Fig 6e/f are different to the analyses used in Fig 3D and 3H. Therefore, on balance I do not feel that the rebuttal adequately addresses the concern raised on this point by the reviewer.

Reply:

We further altered the text on page 9 to highlight the theoretical nature of that analysis. We believe that the text is now very clear on this matter.

3. Concern regarding novelty related to ploidy reduction

I agree with the authors' rebuttal to this point which satisfactorily addresses the concerns of the reviewers. I would highlight that for the non-specialist reader the current version of the manuscript may be challenging with respect to these issues.

Reply:

According to the reviewer #4's comment, we further revised the manuscript with focusing on the clarity of message.

4. Evaluation of relevance of ploidy reduction in greater detail.

The reviewer raises a relevant point, however, I would agree with the authors that while future investigation of this characterisation is of scientific merit it is not required at this time for the fundamental description of the importance of ploidy reduction in carcinogenesis.

Reply:

We appreciate that the reviewer accepts our response about pursuing the experiments proposed by reviewer #3 in the future.

5. Characterisation of ploidy and nuclear organization of hepatocytes

The authors' rebuttal regarding characterisation of this in their previous publication is fair. However my reading of this concern is regarding the characterisation of the cells immediately pre transplantation and post transplantation from the tumors themselves. This analysis would be beneficial in my view. I accept the authors' observation for bias between the YFP and RFP monochrome cells, although appreciate why this is remains untested.

Reply:

As described in our responses to the reviewer #4's major comments, we carefully and clearly proved that bicolored labeling indicates polyploidy, and that loss of a reporter corresponds to ploidy reduction in our previous paper (Matsumoto T, et al. Cell Stem Cell 2020, Figs. 1D, E, 3B and Fig. 6.), and we did not repeat them here. In addition, our previous finding that ploidy reduction in the

livers without transplantation (Matsumoto T, et al. Cell Stem Cell 2020), and the fact that ploidy reduction was observed without ex vivo culture or oncogenic manipulations in naïve hepatocytes, while serially-transplanted cells did not undergo ploidy reduction under the same condition (Figure 2c) supports that ploidy reduction is not associated with experimental manipulations.

Moreover, as also described in our responses to the reviewer #4's major comments, ploidy-reduced hepatocytes are likely to readily re-polyploidize, and liver tumor cells are frequently polyploidized at a variety of frequency in tumors (Bou-Nader M, et al. Gut 2020). Thus, it is unfortunately difficult to deduce tumor origins using ploidy status of established tumors, and the only thing we can confirm with certainty is that bicolored tumors must be derived from polyploid cells.

6. Concerns regarding clumping of cells, especially over time, in vitro prior to sorting

These technical concerns will remain with, as the authors' concede, the inevitable impurity of sorted populations. I appreciate the concerns of the reviewer but overall would agree with the authors that they have suitably (both here and previously) addressed this concern to the best of what is reasonably achievable.

Minor points

1. Quantification of regenerative nodules

Rebuttal accepted

2. Description of experimental technical details

Rebuttal accepted

3. Concern regarding rescue of FAH deficiency experiment (Ext Fig 3)

Personally, I found this an elegant experiment which, as the authors' report, was designed to show that chromosomal aberrations in situ. I accept the authors' rebuttal of this point.

4. Mechanism of reduced ploidy reduction after serial transplantation

The authors highlight the ongoing proliferative potential of serially transplanted hepatocytes in this model based on published data. This point is fair. I would go further than this query made by the reviewer and question whether the effects upon serially transplantation affect tumorigenic potential in ways other than the ability to undergo ploidy reduction as highlighted also by reviewer #1. In answer to the reviewers questions the rebuttal is sufficient, however in order to address concerns regarding experimental validity I remain with concern in this area despite the reassurances provided in the rebuttal.

5. Comparison to baseline transplantation in the serial transplantation studies.

Rebuttal accepted.

6. Presentation of serially transplanted hepatocyte images

Rebuttal accepted

Reply:

We appreciate these thoughtful comments.

Reviewer #1

1 A) Novelty.

I would agree with the authors' rebuttal that the field is currently undecided regarding the role or not of polyploidy with relevance to hepatocellular carcinoma. I would see this as distinct to polyploidy as the liver is an organ where polyploidy is the norm physiologically. I agree with the authors' rebuttal and do not see lack of novelty as a preclusion to publication.

Reply:

We appreciate this comment.

1. B) Alternative explanation for the observations made.

I agree with the reviewers point here which is consistent with my concerns raised in my independent review. While the experimental data is supportive it does not elegantly prove their hypothesis. This issue remains unresolved in the most recent submission. I agree with the previous reviewer that a rescue experiment restoring reductive mitosis in the serially transferred cells would be an excellent way to address this. Whilst I accept the arguments made in rebuttal to this they do not adequately address the concerns raised by the previous reviewer.

Reply:

We agree that a rescue experiment to restore ploidy reduction in serially-transplanted cells would further support our findings that ploidy reduction promotes carcinogenesis. Unfortunately, however, there are no known molecules regulating multipolar reductive mitosis without affecting bipolar mitosis and tumorigenesis regardless of ploidy reduction as described in our previous response to reviewer #1. Thus, we would like to leave this subject to other research.

2. Loading in favor of polyploids in the use of tumor suppressors.

Whilst I see the reviewers point I think that the combination of models using hyperexpression of oncogenes Myc/YAP/Akt in addition to deletion of tumor suppressors p53/PTEN and p16/ARF is a reasonable approach. Obviously the group have attempted additional yet unsuccessful strategies. Therefore on balance I am reassured by the authors' rebuttal and discussion on this issue specifically.

Reply:

We appreciate this comment.

3. Crispr in efficiency related to point 1B above; in serially transplanted hepatocytes.

This is a very good suggestion made by the reviewer which had not occurred to me, and again would be a confounder in the interpretation. This comes back to the validation of the Crispr efficiency. Again this issue would be reconciled by a successful rescue experiment. The authors' rebuttal regarding equal treatment of cells does not equate to equal response with the Crispr method. The equivalent variation between octaploid and tetraploid (measured by TIDE) additionally does not reassure as the key comparison here is to diploids not between polyploids. The rebuttal goes some way to addressing this point but does not fully reassure and the concerns raised relate to point 1B above remain.

Reply:

As described in our response to reviewer #4's comment, transfection of CRISPR/Cas9 efficiently induced p53 and Pten in both diploids and polyploids. Although there was a tendency towards a higher indel frequency in diploids, the difference was minor.

Minor points

1. Rebuttal accepted

2. Rebuttal accepted

3. Rebuttal accepted

4. Rebuttal accepted

5. This is an interesting question but I would side with the authors that the SNP array (to my understanding at least) would not address this point. I do not see addressing this point as critical to the current manuscript.

Reply:

We appreciate these comments.